# ENSEMBLES OF LOW-RANK EXPERT ADAPTERS

**Yinghao Li**[1]**, Vianne Gao**[2]**, Chao Zhang**[2]**, MohamadAli Torkamani**[1]
[1]Amazon Web Service    [2]Amazon.com
{yinghli,gaov,zhanpcha,alitor}@amazon.com

## ABSTRACT

The training and fine-tuning of large language models (LLMs) often involve diverse textual data from multiple sources, which poses challenges due to conflicting gradient directions, hindering optimization and specialization. These challenges can undermine model generalization across tasks, resulting in reduced downstream performance. Recent research suggests that fine-tuning LLMs on carefully selected, task-specific subsets of data can match or even surpass the performance of using the entire dataset. Building on these insights, we propose the Ensembles of Low-Rank Expert Adapters (ELREA) framework to improve the model's capability to handle diverse tasks. ELREA clusters the training instructions based on their gradient directions, representing different areas of expertise and thereby reducing conflicts during optimization. Expert adapters are then trained on these clusters, utilizing the low-rank adaptation (LoRA) technique to ensure training efficiency and model scalability. During inference, ELREA combines predictions from the most relevant expert adapters based on the input data's gradient similarity to the training clusters, ensuring optimal adapter selection for each task. Experiments show that our method outperforms baseline LoRA adapters trained on the full dataset and other ensemble approaches with similar training and inference complexity across a range of domain-specific tasks.

## 1 INTRODUCTION

While general-domain large language models (LLMs) such as GPT-4 (OpenAI, 2022; 2023) and Llama (Touvron et al., 2023) have shown remarkable efficacy in diverse applications, adapting these models through supervised fine-tuning to specific domains or tasks remains indispensable for achieving optimal performance. For example, instruction following requires subtle model adjustments to specialized datasets that the general pre-training corpus alone cannot provide (Ouyang et al., 2022). Significant resources have been invested in constructing varied, high-quality datasets tailored for LLM fine-tuning such as Alpaca (Taori et al., 2023), the Pile (Gao et al., 2021), or Flan (Longpre et al., 2023). These efforts have fueled the development of specialized models that address complex tasks across fields such as medical diagnostics (Singhal et al., 2023), financial analytics (Yang et al., 2023), and scientific decision-making (Zhang et al., 2024b), or to provide reasoning to their results (Wei et al., 2022), which were tasks once deemed challenging for automated systems.

Nonetheless, fine-tuning LLMs on a comprehensive dataset frequently encounters the issue of conflicting gradient directions from varied training data points (Wang et al., 2021; Xia et al., 2024; Chen et al., 2024). This phenomenon complicates the update process of models, potentially leading to suboptimal performance. Wang et al. (2023d) demonstrate that mixing diverse instructional datasets can sometimes result in less than ideal outcomes compared to fine-tuning on a carefully selected subset of the data that directly addresses the task at hand. To enhance the relevance of training data to specific tasks, Xie et al. (2023) have proposed methods like importance resampling, which aligns the training dataset more closely with the target task distribution. Another innovative approach proposed by Xia et al. (2024), termed targeted instruction tuning, involves selecting a small percentage (about 5%) of training data that most significantly influences task performance based on the average gradients of tokens. This method has shown promise, achieving comparable or superior results to traditional full dataset fine-tuning across various tasks. In addition, Xia et al. (2024) also present better outcomes in selecting data points based on the gradient norm than sentence embeddings.

Despite these advancements, current data selection techniques for fine-tuning are predominantly target-driven, relying heavily on specific features of the target task (*e.g.*, n-gram frequency, example answer embedding, gradient direction) to guide the selection process. This requirement for task-specific data features imposes significant limitations when adapting LLMs to new or emerging tasks, especially when relevant training data or features are unavailable.

To address these challenges, we propose a novel framework, Ensembles of Low-Rank Expert Adapters (ELREA), which leverages Low-Rank Adaptation (LoRA; Hu et al., 2022; Dettmers et al., 2023) to create multiple expert adapters. These adapters are trained independently on data groups with similar gradient directions and their predictions are assembled during inference based on the gradient features of the input. Specifically, ELREA begins by fine-tuning a base adapter on the full dataset to capture a wide range of general knowledge. We then evaluate and cluster the gradients of individual data points relative to their influence on the base adapter, organizing them into similarly sized groups. On each cluster we continue training a specialized LoRA expert that is initialized from the base adapter, allowing the training process to maintain a comparable computational burden to that of a single adapter trained on the entire dataset. During inference, the expert adapters collaboratively determine the output by dynamically weighting the adapters according to their alignment with the clusters' gradient profile. Compared with conventional Deep Ensembles, such calculation could be conducted only once at the beginning in the recurrent generation process and re-used in subsequent passes, causing minimal computational overhead while achieving stronger performance (Lakshminarayanan et al., 2017; Havasi et al., 2021; Wang et al., 2023a). Unlike previous methods, ELREA focuses on the task-agnostic setup, *i.e.*, a one-time training effort without the need for additional task-specific validation data, making it more suitable for real-world deployment of LLMs.

In summary, our contributions are threefold:

- We introduce Ensembles of Low-Rank Expert Adapters (ELREA), a framework that integrates efficient parameter adaptation techniques into an ensemble model to address conflicting gradient directions in LLM fine-tuning.
- By combining gradient features with clustering methods, we create expert adapters specialized for different gradient profiles, enabling the model to adapt to diverse tasks without relying on task-specific data features or validation data points.
- We demonstrate that ELREA outperforms baseline LoRA adapters trained on the full dataset across various domain-specific applications, as well as other Mixture of Experts (MoE) and self-consistency methods.

## 2 PRELIMINARIES

### 2.1 LANGUAGE MODELS AND PARAMETER-EFFICIENT FINE-TUNING

Decoder-only LMs, pioneered by GPT (Radford et al., 2018), are built upon the decoder component of the Transformer architecture (Vaswani et al., 2017) and are among the most prevalent and thoroughly examined language models today. A pre-trained LM, denoted as $\mathcal{M}$, learns the language patterns on extensive text corpora $\mathcal{D}_{\text{pre-train}}$ through an unsupervised next-token-prediction (NTP) objective, which minimized the negative log likelihood (NLL) of a subsequent token $x_t$ in a length-$T$ sequence $\boldsymbol{x} \in \mathbb{V}^T$ consisting tokens from the vocabulary $\mathbb{V}$ based on the preceding context $\boldsymbol{x}_{<t}$:

$$\mathcal{L}_{\text{NTP}}(\boldsymbol{x}) = -\sum_{t=1}^{T} \log P(x_t | \boldsymbol{x}_{<t}; \boldsymbol{\theta}_{\mathcal{M}}), \tag{1}$$

where $\boldsymbol{\theta}_{\mathcal{M}}$ are the network parameters of the LM. Originally designed for text completion, the pre-trained LMs have been enhanced with instruction-following or task-specific capabilities through targeted fine-tuning (Ouyang et al., 2022; OpenAI, 2022; 2023), expanding their utility across diverse applications. The fine-tuning process frequently adopts the NTP objective, utilizing a smaller, specialized fine-tuning dataset $\mathcal{D}_{\text{ft}}$ that consists of instruction-response pairs $\boldsymbol{x}_{\text{ft}} = (\boldsymbol{x}_{\text{instr}}, \boldsymbol{x}_{\text{resp}})$.

Full-parameter fine-tuning of high-performing LMs, which involves calculating $\nabla_{\boldsymbol{\theta}_{\mathcal{M}}} \mathcal{L}_{\text{NTP}}(\boldsymbol{x})$ and updating $\boldsymbol{\theta}_{\mathcal{M}}$ accordingly, is often impractical due to computational constraints arising from their vast number of parameters. To address this issue, parameter-efficient fine-tuning (PEFT) techniques have been developed (Houlsby et al., 2019; Li & Liang, 2021; He et al., 2022), with LoRA being a prominent example. LoRA introduces adapter $\boldsymbol{\theta}_{\mathcal{Q}}$ into the LM's linear layers whose weight

matrices are, for example, $\boldsymbol{W}_i \in \mathbb{R}^{d_{\text{model}} \times d_{\text{model}}}$, where $i$ is the layer index and $d_{\text{model}}$ is the model dimensionality as defined in (Vaswani et al., 2017). LoRA approximates the weight adjustments during fine-tuning using a low-rank decomposition $\Delta \boldsymbol{W}_i \approx \boldsymbol{A}_i \boldsymbol{B}_i^{\mathsf{T}}$. Here, $\boldsymbol{A}_i, \boldsymbol{B}_i \in \mathbb{R}^{d_{\text{model}} \times r}$ are rank-$r$ adapter matrices with $r \ll d_{\text{model}}$. During fine-tuning, the original weight matrices $\boldsymbol{W}_i$ remain frozen, and only the adapter parameters $\boldsymbol{\theta}_{\mathcal{Q}} \triangleq \bigcup_i \{\boldsymbol{A}_i, \boldsymbol{B}_i\}$ are updated to minimize the NLL loss: $\min_{\boldsymbol{\theta}_{\mathcal{Q}}} \mathcal{L}_{\text{NTP}}(\boldsymbol{x}; \boldsymbol{\theta}_{\mathcal{M}} + \boldsymbol{\theta}_{\mathcal{Q}})$. PEFT significantly reduces the computational demands of fine-tuning by limiting gradient calculations to a smaller set of parameters.

## 2.2 GRADIENT FEATURE CALCULATION AND DATA SELECTION

Originally introduced by Pruthi et al. (2020) to estimate the impact of individual training examples on model performance, gradient-based data selection has been further applied to training data selection (Gou et al., 2023; Xia et al., 2024; Pan et al., 2024; Liu et al., 2024c; Yang et al., 2024). Unlike methods based on surface-form textual features—which utilize token statistics or sentence embeddings as selection criteria (Reimers & Gurevych, 2019; Xie et al., 2023), this approach employs parameter gradients $\nabla_{\boldsymbol{\theta}}$ instead. Specifically, when fine-tuning a LoRA adapter $\mathcal{Q}$ using stochastic gradient descent (SGD), the gradient feature $\boldsymbol{g}(\boldsymbol{x})$ for each sequence $\boldsymbol{x}$ can be computed as

$$\boldsymbol{g}(\boldsymbol{x}) \in \mathbb{R}^{|\boldsymbol{\theta}_{\mathcal{Q}}|} = \text{flatten}\left(\nabla_{\boldsymbol{\theta}_{\mathcal{Q}}} \mathcal{L}_{\text{NTP}}(\boldsymbol{x})\right). \tag{2}$$

$\text{flatten}(\cdot)$ denotes the operation that reshapes matrices into vectors and concatenates them. Using this expression, we derive the trajectory influence of a training data point $\boldsymbol{x}_{\text{ft}} \in \mathcal{D}_{\text{ft}}$, quantified by the inner product between its gradient feature and that of a task-specific validation data point $\boldsymbol{x}_{\text{valid}}$. This inner product is then accumulated across training epochs $e$, each weighted by the average learning rate $\eta^{(e)}$ for that epoch: $\sum_{e=1}^{E} \eta^{(e)} \langle \boldsymbol{g}(\boldsymbol{x}_{\text{ft}}), \boldsymbol{g}(\boldsymbol{x}_{\text{valid}}) \rangle$. By leveraging this formulation and adapting it to the Adam optimizer (§ 3.2), Xia et al. (2024) demonstrate the efficacy of selecting a subset of training data with the highest influence scores for task-specific fine-tuning, achieving performance comparable to that obtained using the complete training dataset.

## 2.3 MIXTURE OF EXPERTS AND ENSEMBLES

Mixture of Experts (MoE) is an architecture that combines multiple expert models or network modules with a gating network (Szymanski & Lemmon, 1993; Jordan & Jacobs, 1994). In the context of LLMs, MoE was first adopted by Shen et al. (2023) for instruction-tuning and by Jiang et al. (2024) for LLM pre-training to reduce inference costs while achieving performance comparable to dense networks. This idea has been further developed in subsequent works (Zhu et al., 2024; Dai et al., 2024; Xue et al., 2024).

Upon receiving an input, the MoE's gating network routes it to the relevant experts, which could be an entire feed-forward Transformer block (Jiang et al., 2024) or a fine-tuned LoRA adapter (Dou et al., 2023; Wu et al., 2024) for LMs. Routing could be either dense or sparse, depending on the fraction of the total experts are activated. The selected experts process the input and provide their outputs, which are aggregated at the end of the layer or block, typically through weighted averaging, to produce the final result. This dynamic and selective activation of experts ensures efficient computation and resource utilization. Mathematically, the output of a mixture of $M$ experts can be expressed as:

$$\mathcal{F}(\boldsymbol{x}) = \sum_{m=1}^{M} \lambda_m(\boldsymbol{x}) \mathcal{E}_m(\boldsymbol{x}); \quad \sum_{m=1}^{M} \lambda_m(\boldsymbol{x}) = 1, \; \left| \{m | \lambda_m(\boldsymbol{x}) \neq 0\}_{m=1}^{M} \right| \leq M, \tag{3}$$

where $\mathcal{E}_m$ is an expert model, and $0 \leq \lambda_m \leq 1$ is its weight predicted by the gating network. Here we extend the definition of $\boldsymbol{x}$ to any kind of layer input.

On the other hand, Deep Ensembles utilize a collection of multiple models with identical architecture that are trained independently with different parameter initializations (Lakshminarayanan et al., 2017; Gleave & Irving, 2022) . During inference, the last-layer predictions of these models, which could be either pre-activation logits or post-activation probabilities, are averaged to improve the overall performance. Suppose we have $N$ models $\{\mathcal{M}_n\}_{n=1}^{N}$ in the ensemble, the output would be:

$$\mathcal{M}_{\text{ens}}(\boldsymbol{x}) = \frac{1}{N} \sum_{n=1}^{N} \mathcal{M}_n(\boldsymbol{x}). \tag{4}$$

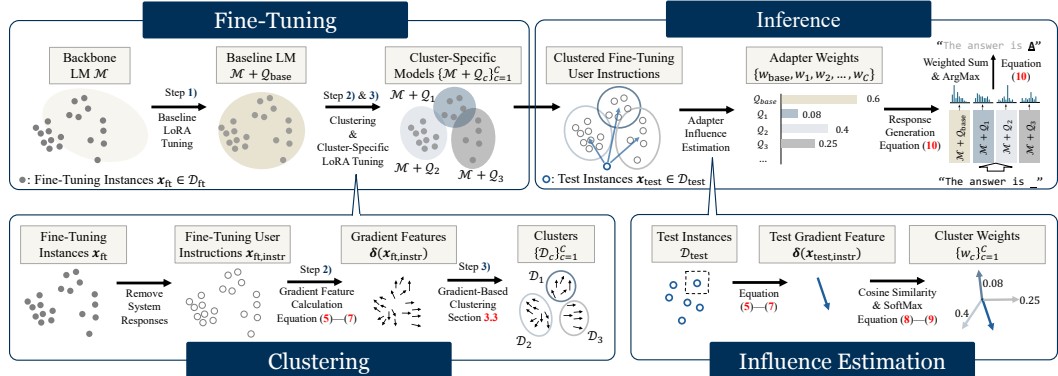

Figure 1: The pipeline of ELREA for fine-tuning and inference. The data points (solid and hollow circles) do not necessarily have a geometric correspondence to their gradient directions (arrows).

The major differences between MoE and Deep Ensembles are two-fold: 1) MoE uses *trainable* gating networks for model selection, while Deep Ensembles average uniform or pre-defined weights; 2) MoE conducts output aggregation within layers or blocks, while Deep Ensembles do so at the end of the model. Although MoE can achieve finer-grained routing and potentially superior performance with careful design, Deep Ensembles, as both theoretically and empirically shown, remain the top approach for robustly improving model performance in value prediction and uncertainty estimation, albeit at the cost of reduced efficiency (Lakshminarayanan et al., 2017; Garipov et al., 2018; Fort et al., 2019; Fang et al., 2023; Pitis et al., 2023; Li et al., 2024b). For a more detailed discussion on MoE and Deep Ensembles in the context of LLMs, please refer to appendix A.

## 3 METHOD

In this section, we introduce the pipeline of ELREA, designed to enhance the fine-tuning of LLMs for improved downstream tasks by leveraging a mixture of LoRA experts in a Deep Ensembles framework. The pipeline, shown in Figure 1, consists of three main steps: 1) full-data adapter tuning, 2) gradient calculation, and 3) clustering and per-cluster fine-tuning. During inference, we estimate the similarity between the gradient of test instructions and the cluster instances to determine the influence of each cluster on the final prediction. The details of each step are elaborated below.

### 3.1 FULL-DATA ADAPTER TUNING

The first step involves fine-tuning a base LoRA adapter $\mathcal{Q}_{\text{base}}$ from the backbone language model $\mathcal{M}$ on the entire fine-tuning dataset $\mathcal{D}_{\text{ft}}$ for $E$ epochs using the NTP objective (equation 1). This process captures a broader spectrum of general and task-specific knowledge and enhances the model's basic instruction-following abilities. The adapted model checkpoints $\{\mathcal{M} + \mathcal{Q}_{\text{base}}^{(e)}\}_{e=1}^{E}$, where $\mathcal{Q}_{\text{base}}^{(e)}$ denotes the adapter checkpoint at the end of training epoch $e$, along with the corresponding optimizer states, provide the necessary parameters to calculate the gradient features (Xia et al., 2024).[1]

### 3.2 GRADIENT CALCULATION

With Adam optimizer (Kingma & Ba, 2015), which is the most adopted for LM fine-tuning, the gradient feature $g(x)$ for each sequence $x$ is extended from equation 2 to consider the 1st and 2nd order momentum terms with decay rates $\beta_1$ and $\beta_2$, as derived by Xia et al. (2024):

$$g_{\text{Adam}}^{(t)}(x) = \eta^{(t)} \cdot m^{(t)}/(\sqrt{v^{(t)}} + \epsilon);$$
$$m^{(t)} = (\beta_1 m^{(t-1)} + (1 - \beta_1)g)/(1 - \beta_1^t); \quad v^{(t)} = (\beta_2 v^{(t-1)} + (1 - \beta_2)g^2)/(1 - \beta_2^t),$$

$\qquad\qquad\qquad\qquad\qquad\qquad\qquad\qquad\qquad\qquad\qquad\qquad\qquad\qquad (5)$

---

[1] Here we extend the definition of the operator "+" between the backbone model and an adapter to denote the addition of the weights of the corresponding network layers (Hu et al., 2022).

where $t$ is the current training step and $\epsilon$ is a small constant to prevent division by zero. Each training instance $\boldsymbol{x}_{\text{ft}} \in \mathcal{D}_{\text{ft}}$ is then associated with $E$ gradients $\{\boldsymbol{g}_{\text{Adam}}^{(e)}(\boldsymbol{x}_{\text{ft}})\}_{e=1}^{E}$, each with the dimensionality of the number of total parameters in the adapter $|\boldsymbol{\theta}_{\mathcal{Q}}|$.[2]

Although $|\boldsymbol{\theta}_{\mathcal{Q}}| \ll |\boldsymbol{\theta}_{\mathcal{M}}|$, it is still at a million level scale, which is too large for efficient clustering or similarity computation. Therefore, we follow Xia et al. (2024) and apply random projection (Kanerva et al., 2000), which is derived from the Johnson-Lindenstrauss lemma (Johnson & Lindenstrauss, 1984) stating that sufficiently high-dimensional data points can be projected into lower-dimensional space while approximately preserves pairwise distances between the points, to reduce the dimensionality of the gradient features to $d_{\text{proj}} \ll |\boldsymbol{\theta}_{\mathcal{Q}}|$.

$$\boldsymbol{g}_{\text{Adam}}' = \boldsymbol{R}\boldsymbol{g}_{\text{Adam}}; \quad \boldsymbol{R} \in \{-1, 1\}^{d_{\text{proj}} \times |\boldsymbol{\theta}_{\mathcal{Q}}|}; \ R_{ij} \sim \mathcal{U}(\{-1, 1\}). \tag{6}$$

For gradient feature clustering, we first average the gradient features of each instance across all epochs to obtain a single representative feature vector, which is then normalized and projected into a $(d_{\text{proj}} - 1)$-dimensional hyper-sphere:

$$\boldsymbol{\delta}(\boldsymbol{x}) = \frac{\boldsymbol{\delta}'(\boldsymbol{x})}{\|\boldsymbol{\delta}'(\boldsymbol{x})\|}; \quad \boldsymbol{\delta}'(\boldsymbol{x}) = \frac{1}{E}\sum_{e=1}^{E} \boldsymbol{g}_{\text{Adam}}'^{(e)}(\boldsymbol{x}) \tag{7}$$

as we are only interested in the gradient directions rather than their magnitudes.

ELREA is developed under the assumption that the test distribution is entirely unknown during fine-tuning. Therefore, for both fine-tuning and test instances, we only consider the gradient of the instruction (*i.e.* user-input) tokens $\boldsymbol{x}_{\text{instr}}$ (§ 2.1), excluding the expected system responses even if they are provided in the training data, which is different from Xia et al. (2024) who construct the gradients based only on the expected model answers.

### 3.3 CLUSTERING AND PER-CLUSTER FINE-TUNING

We then cluster the training gradient features $\{\boldsymbol{\delta}(\boldsymbol{x}_{\text{ft, instr}}) | \boldsymbol{x}_{\text{ft, instr}} \in \mathcal{D}_{\text{ft}}\}$ into $K$ clusters using the BIRCH algorithm (Zhang et al., 1996). The BIRCH algorithm is well-suited for large, high-dimensional datasets and demonstrates robustness against outliers. To reduce computational demands, we randomly select 5,000 data points from $\mathcal{D}_{\text{ft}}$ for model fitting. This sample size adequately represents the feature distribution, and we use the resulting model to cluster all gradient features. Preliminary experiments show that the clustering algorithm is robust, *i.e.*, it consistently produces identical or similar clusters when different random seeds are used. As BIRCH does not ensure balanced clusters, we reapply it to clusters exceeding five times the size of the smallest cluster. We iterate this process up to three times, each iteration targeting fewer clusters. Initially targeting 5 clusters, this method typically yields between 8 (after two iterations) and 10 (after three iterations) training clusters $\{\mathcal{D}_c\}_{c=1}^{C}$, where $C$ denotes the final number of clusters.

Within each cluster $\mathcal{D}_c$, we proceed with LoRA fine-tuning from the base checkpoint $\mathcal{Q}_{\text{base}}^{(E)}$, extending for several more epochs at a reduced learning rate utilizing the same NTP objective. This results in a collection of $C$ LoRA expert adapters $\{\mathcal{Q}_c\}_{c=1}^{C}$. Theoretically, each cluster contains training instances with similar gradient directions, which likely exert analogous effects on the model's behavior. Fine-tuning with clustered data aims to direct the model towards a more precise update path, thereby potentially enhancing the model's (*i.e.*, $\mathcal{M} + \mathcal{Q}_c$) performance on specific task types which are unidentified during fine-tuning.

### 3.4 ROUTING AND INFERENCE

To route an input instruction to appropriate expert adapters, we calculate the cosine similarity between the gradient of the instruction $\boldsymbol{\delta}_{\text{test}} \triangleq \boldsymbol{\delta}(\boldsymbol{x}_{\text{test, instr}})$ and the centroid of the gradients within each cluster $\bar{\boldsymbol{\delta}}_c' = \frac{1}{|\mathcal{D}_c|}\sum_{\boldsymbol{x}_i \in \mathcal{D}_c} \boldsymbol{\delta}(\boldsymbol{x}_{i,\text{instr}})$. The normalized form of $\bar{\boldsymbol{\delta}}_c$ is given by:

$$\bar{\boldsymbol{\delta}}_c = \frac{\bar{\boldsymbol{\delta}}_c'}{\|\bar{\boldsymbol{\delta}}_c'\|}; \quad \bar{\boldsymbol{\delta}}_c' = \frac{1}{|\mathcal{D}_c|}\sum_{\boldsymbol{x}_i \in \mathcal{D}_c} \boldsymbol{\delta}(\boldsymbol{x}_{i,\text{instr}}). \tag{8}$$

---

[2]We use the same rank for all adapters, so we do not emphasize the difference of adapters here.

Here, the cosine similarity simply becomes the inner product of these two normalized vectors: $\cos(\boldsymbol{\delta}_{\text{test}}, \bar{\boldsymbol{\delta}}_c) = \langle \boldsymbol{\delta}_t, \bar{\boldsymbol{\delta}}_c \rangle$. When the projection dimensionality $d_{\text{proj}}$ is high, the similarity may suffer from the curse of dimensionality, where the gaps between the similarities to different cluster centroids may become too small. To address this issue, we standardize the cosine similarities across clusters before employing a SoftMax function on the standardized similarities $\cos'(\boldsymbol{\delta}_{\text{test}}, \bar{\boldsymbol{\delta}}_c)$ across clusters to determine their respective weights:

$$w_c = \frac{\exp(\cos'(\boldsymbol{\delta}_{\text{test}}, \bar{\boldsymbol{\delta}}_c))}{\sum_{c'=1}^{C} \exp(\cos'(\boldsymbol{\delta}_{\text{test}}, \bar{\boldsymbol{\delta}}_{c'}))}; \quad \cos'(\boldsymbol{\delta}_{\text{test}}, \bar{\boldsymbol{\delta}}_c) = \frac{\cos(\boldsymbol{\delta}_{\text{test}}, \bar{\boldsymbol{\delta}}_c) - \mu_{\text{test}}}{\sigma_{\text{test}}}, \quad (9)$$

where $\mu_{\text{test}}$ and $\sigma_{\text{test}}$ are the mean and standard deviation of the cosine similarities across clusters.

Besides the cluster-specific adapters $\{\mathcal{Q}_c\}_{c=1}^{C}$, we also incorporate the base adapter $\mathcal{Q}_{\text{base}}$ during inference to leverage the general knowledge captured from the entire dataset. This is particularly crucial when the test instruction diverges significantly from all training instances, indicated by $\max_c\{\cos(\boldsymbol{\delta}_{\text{test}}, \bar{\boldsymbol{\delta}}_c)\} < \tau$, where $\tau$ is some threshold. We quantify the influence of the base adapter as $w_{\text{base}} = 1 - \max_c\{\cos(\boldsymbol{\delta}_{\text{test}}, \bar{\boldsymbol{\delta}}_c)\}$. Therefore, we assemble $C + 1$ adapters during inference, with the final prediction for the next token being the ArgMax of the weighted sum of output logits from each adapter:

$$\hat{x}_t = \arg\max_{x_t} \left( w_{\text{base}}(\mathcal{M} + \mathcal{Q}_{\text{base}})(x_t|\boldsymbol{x}_{<t}) + \sum_{c=1}^{C} w_c(\mathcal{M} + \mathcal{Q}_c)(x_t|\boldsymbol{x}_{<t}) \right), \quad (10)$$

which is a combination of equation 3 and equation 4. $x_t$ is categorical, while $\mathcal{M}(x_t|\boldsymbol{x}_{<t})$ denotes the output pre-activation logit of categorical token $x_t$ given the context tokens $\boldsymbol{x}_{<t}$ from the language model $\mathcal{M}$. In equation 10 we get $\hat{x}_t$, we append it to the context tokens $\boldsymbol{x}_{<t+1} = (\boldsymbol{x}_{<t}, \hat{x}_t)$ for *all* adapters in the ensemble and repeat the process until the end of the sequence is reached. As we are not dealing with probabilities here, the weights do not need to sum to 1, *i.e.* $w_{\text{base}} + \sum_{c=1}^{C} w_c \neq 1$.

Unlike the LoRA MoE approaches (§ 2.3), which utilizes a gating network for layer-wise routing with predictions aggregated post-layer, ELREA resembles Deep Ensembles in its routing and aggregation strategy but uses LoRA adapters as ensemble components, and hence the name.

## 4 EXPERIMENTAL SETUP

**Datasets** We conducted experiments across two distinct evaluation categories: 1) general language understanding and reasoning, and 2) mathematical reasoning. For the first category, following Xia et al. (2024), we employ Flan V2 (Longpre et al., 2023), CoT (Wei et al., 2022), Dolly-15k (Conover et al., 2023), and OpenAssistant Conversations (Köpf et al., 2023) for fine-tuning, and MMLU (Hendrycks et al., 2021a) and BIG-bench Hard (BBH; bench authors, 2023; Suzgun et al., 2023) to test model performance. The training and test datasets have no distribution overlap, making this setup suitable for evaluating the model's generalization capabilities. For the mathematical reasoning category, we develop the MATH-Combined dataset by integrating existing resources including GSM8k (Cobbe et al., 2021), MathQA (Amini et al., 2019), SVAMP (Patel et al., 2021), and MATH (Hendrycks et al., 2021b) into a uniform format analogous to MATH. MATH-Combined utilizes in-domain test points, offering insights into selecting task-specific data for effective fine-tuning. Please refer to appendix B for dataset details and processing; and Table 4 for the statistics.

**Model and Fine-Tuning** Our primary experiments involve fine-tuning the Gemma-2b model (Gemma Team, 2024b), specifically `gemma-1.1-2b-it`[3], by applying rank-8 LoRA adapters to all linear layers, modifying about 0.39% of the total model parameters. For both dataset categories, we fine-tune the base adapter $\mathcal{Q}_{\text{base}}$ for 2 epochs using the Adam optimizer, with an initial learning rate of $5 \times 10^{-5}$ that linearly decays to zero. Cluster-wise adapters $\mathcal{Q}_c$ are initialized from $\mathcal{Q}_{\text{base}}$ and fine-tuned for the same duration with a slightly reduced learning rate of $2 \times 10^{-5}$. These hyperparameters are fixed since we assume no access to additional task-specific validation data. The maximum token sequence length during training is 2,048, with a batch size equals to 16 sequences distributed across the GPUs. Following Xia et al. (2024), we set the gradient projection dimensionality for clustering $d_{\text{proj}}$ to 8,192, which we show leads to the best model performance. Please refer to appendices C and D for additional details.

---

[3]Available at https://huggingface.co/google/gemma-1.1-2b-it.

**Inference and Evaluation**   Since the test set is out of the fine-tuning distribution for the general reasoning and understanding category, we use up to three in-context examples from the validation subset of BBH and five from MMLU. For the mathematical reasoning category, we employ a zero-shot setup. During inference, we limit the maximum instruction sequence length to $1,200$ tokens and the response length to $848$ tokens for MATH-Combined and BBH. For MMLU, the instruction length is increased to $1,800$ tokens and the response length to $248$ tokens. We reduce the number of in-context examples until the instruction length falls within the specified limits. We employ greedy decoding at zero temperature and maximize the batch size feasible under operational constraints. For MATH-Combined, we leverage existing code from Hendrycks et al. (2021b) for parsing results and assessing accuracy.[4] For MMLU and BBH, we develop regular expressions to parse outputs and calculate exact-match accuracy metrics. It is worth noting that although we use Gemma-2b as the backbone model $\mathcal{M}$, we do not adhere to the experimental setup or evaluation protocol described in (Gemma Team, 2024b). Consequently, our reported results may differ from theirs.

**Baselines**   The baseline model, $\mathcal{M} + \mathcal{Q}_{\text{base}}$, is fine-tuned on the entire dataset, serving as a general reference point for comparison. $\mathcal{M} + \mathcal{Q}_{\text{dataset}}$ represents adapters fine-tuned and applied separately to each subset of MATH-Combined. For the backbone-only category, we directly evaluate the performance of the backbone model $\mathcal{M}$. To compare with the MoE setup, we include three baselines: MoE Routing, MoE Merging, and MoLE. MoE Routing implements layer-level routing using the same weights as ELREA. MoE Merging averages the expert network parameters based on the expert weights before processing the input. Mixture of LoRA Experts (MoLE, Wu et al., 2024) applies a layer-wise gating function to dynamically predict expert weights based on the layer inputs. From the ensembling family, we consider Self-Consistency and LoRA Ensembles. Self-Consistency (Wang et al., 2023b) uses $\mathcal{M} + \mathcal{Q}_{\text{base}}$ as the base model, performing five inference passes per instance with a temperature of 1. The final prediction is determined through majority voting. LoRA Ensembles (Wang et al., 2023a) independently fine-tunes three additional adapters, aside from $\mathcal{Q}_{\text{base}}$, under the same setup and averages predictions across all four models. To investigate the efficacy of gradient-based features, we have the Instruction Embedding baseline, which substitutes instruction gradients with sentence embeddings from a pre-trained model for data clustering and instance routing.

Additionally, we include Random Cluster and Uniform Weights as ablation study baselines. Random Cluster maintains the same cluster numbers and sizes as ELREA but assigns cluster members randomly from the $\mathcal{D}_{\text{ft}}$, which preserves the distribution characteristics of $\mathcal{D}_{\text{ft}}$ and positions it as an approximate Deep Ensembles baseline with equivalent training effort to ELREA. On the other hand, Uniform Weights assigns equal weights to all clusters to verify the effectiveness of the cluster-wise adapter routing mechanism. Please refer to appendix E for baseline details.

## 5   RESULTS AND DISCUSSION

**Main Results**   Table 1 presents the test set accuracy across various MATH-Combined subsets, along with the micro-averaged results. ELREA consistently outperforms baseline methods on most sub-datasets by an observable margin, with only occasional dips in performance. On average, EL-REA achieves performance gains of 9.67% and 3.56% over $\mathcal{M} + \mathcal{Q}_{\text{base}}$ at ranks $r = 8$ and $r = 64$, respectively, without leveraging additional training data or external knowledge sources. Table 2 further highlights the robustness of ELREA in general language understanding and reasoning tasks, even under test conditions that diverge from those used during fine-tuning. This finding aligns with the results reported by Xia et al. (2024). A comparison between $\mathcal{M} + \mathcal{Q}_{\text{dataset}}$ and $\mathcal{M} + \mathcal{Q}_{\text{base}}$ reveals that the former does not consistently outperform the latter. This observation suggests that a generalized approach to knowledge extraction across similar tasks (as illustrated in Figure 4) can sometimes be more effective than relying solely on dataset-specific expertise.

The MoE Routing and Merging frameworks, despite relying on pre-computed routing weights, still exhibit improvements over the baseline, which can be attributed to the ensemble effect of the experts. In contrast, the MoLE baseline, which employs a trainable router, consistently underperforms compared to $\mathcal{M} + \mathcal{Q}_{\text{base}}$. We hypothesize that the presence of multiple LoRA experts, each applied to a broad range of linear layers (appendix C), may lead to a large and complex scope of routing functions that is challenging to optimize. Consequently, the system likely converges to a suboptimal

---

[4]Available at `https://github.com/hendrycks/math`.

Table 1: Comparison of test set accuracies (in %) across various MATH-Combined subsets, along with the **micro**-average. Gray rows indicate the primary baseline; blue rows highlight ELREA.

| LoRA Rank | Methods | MATH | GSM8k | SVAMP | MathQA | Average[a] |
|---|---|---|---|---|---|---|
| | **Gemma-2b** | | | | | |
| | $\mathcal{M} + \mathcal{Q}_{base}$ | 9.2 | 22.1 | 46.07 | 16.83 | 18.61 |
| | $\mathcal{M} + \mathcal{Q}_{dataset}$ | 7.3 | 25.7 | 45.00 | 16.73 | 19.01 (+ 0.40) |
| | MoE Routing | 9.2 | 22.7 | 48.21 | 16.23 | 18.79 (+ 0.18) |
| | MoE Merging | 9.1 | 23.1 | 48.21 | 15.73 | 18.73 (+ 0.12) |
| | MoLE | 8.8 | 21.6 | 46.43 | 15.53 | 17.99 (− 0.62) |
| $r = 8$ | LoRA Ensembles | 9.3 | 24.7 | 47.50 | 16.73 | 19.55 (+ 0.94) |
| | Self-Consistency | 5.9 | 14.3 | 44.64 | 10.32 | 13.12 (− 5.49) |
| | Instruction Embedding | 9.8 | 24.1 | 46.79 | 16.83 | 19.46 (+ 0.85) |
| | **ELREA** | 9.1 | 25.9 | 49.64 | 18.04 | **20.41 (+ 1.80)** |
| | Random Cluster | 9.1 | 25.1 | 48.21 | 18.84 | 20.30 (+ 1.69) |
| | Uniform Weights | 9.6 | 25.2 | 47.50 | 18.04 | 20.16 (+ 1.55) |
| | $\mathcal{M} + \mathcal{Q}_{base}$ | 10.8 | 32.7 | 55.36 | 27.56 | 26.39 |
| | $\mathcal{M} + \mathcal{Q}_{dataset}$ | 10.8 | 33.0 | 52.14 | 27.66 | 26.24 (− 0.15) |
| | MoE Routing | 11.7 | 31.9 | 60.36 | 26.95 | 26.66 (+ 0.27) |
| | MoE Merging | 11.4 | 32.0 | 60.36 | 26.85 | 26.57 (+ 0.18) |
| | MoLE | 10.7 | 31.7 | 56.07 | 25.35 | 25.49 (− 0.90) |
| $r = 64$ | LoRA Ensembles | 12.1 | 31.8 | 60.00 | 28.06 | 27.06 (+ 0.67) |
| | Self-Consistency | 9.3 | 28.5 | 60.36 | 21.84 | 23.34 (− 3.05) |
| | Instruction Embedding | 11.2 | 31.7 | 60.71 | 28.46 | 26.94 (+ 0.55) |
| | **ELREA** | 12.5 | 32.6 | 57.86 | 28.36 | **27.33 (+ 0.94)** |
| | Random Cluster | 11.5 | 32.8 | 59.64 | 27.05 | 26.87 (+ 0.48) |
| | Uniform Weights | 11.4 | 31.5 | 60.00 | 27.15 | 26.48 (+ 0.24) |
| | **Gemma2-9b** | | | | | |
| $r = 8$ | $\mathcal{M} + \mathcal{Q}_{base}$ | 37.9 | 78.7 | 84.64 | 50.30 | 58.11 |
| | **ELREA** | 37.4 | 78.6 | 86.43 | 52.00 | 58.60 (+ 0.49) |
| $r = 64$ | $\mathcal{M} + \mathcal{Q}_{base}$ | 37.4 | 81.3 | 86.07 | 57.82 | 61.17 |
| | **ELREA** | 36.8 | 80.7 | 87.50 | 59.32 | 61.38 (+ 0.21) |

[a] The number in parentheses indicates the improvement over the corresponding baseline $\mathcal{M} + \mathcal{Q}_{base}$.

Table 2: Comparison of test set exact-match accuracy (in %) on BBH and MMLU, and the **macro**-averaged result. We also include the backbone $\mathcal{M}$ for reference.

| LoRA Rank | Methods | BBH | MMLU | Macro Average |
|---|---|---|---|---|
| N/A | Backbone $\mathcal{M}$[a] | 9.17 | 9.12 | 9.15 |
| | $\mathcal{M} + \mathcal{Q}_{base}$ | 27.20 | 33.73 | 30.47 |
| | MoE Routing | 27.46 (+ 0.26) | 34.21 (+ 0.48) | 30.84 (+ 0.37) |
| | MoE Merging | 27.13 (− 0.07) | 33.98 (+ 0.25) | 30.36 (+ 0.09) |
| | MoLE | 26.40 (− 0.80) | 34.19 (+ 0.46) | 30.30 (− 0.17) |
| $r = 8$ | Self-Consistency | 23.74 (− 3.46) | 32.88 (− 0.85) | 28.31 (− 2.16) |
| | Instruction Embedding | 26.50 (− 0.70) | 34.76 (+ 1.03) | 30.63 (+ 0.16) |
| | **ELREA** | **28.03 (+ 0.83)** | **34.84 (+ 1.11)** | **31.44 (+ 0.97)** |
| | Random Cluster | 27.72 (+ 0.52) | 34.56 (+ 0.83) | 31.14 (+ 0.67) |
| | Uniform Weights | 27.32 (+ 0.12) | 34.33 (+ 0.60) | 30.83 (+ 0.36) |

[a] A large portion of responses are unparsable, leading to an accuracy lower than random guess.

solution, overfitting the training data while sacrificing model generalization. A more sophisticated network design or a refined training strategy may be necessary for MoLE to achieve better results.

Conversely, the classical LoRA Ensembles setup, despite its higher computational cost, demonstrates robustness by consistently outperforming $\mathcal{M} + \mathcal{Q}_{base}$. This . These findings align with our discussion in § 2.3 and underscore the effectiveness of ELREA' ensemble approach. The Self-Consistency method, however, delivers poorer results due to significant variance in outcomes across

Table 3: The performance of ELREA with different clustering methods. The results use Gemma-2b backbone, LoRA rank $r = 64$, and number of clusters $C = 10$.

| Methods | MATH | GSM8k | SVAMP | MathQA | Average |
|---|---|---|---|---|---|
| **ELREA** | 12.5 | 32.6 | 57.86 | 28.36 | 27.33 |
| BIRCH w/ 256-d PCA | 10.3 | 32.1 | 60.00 | 26.95 | 26.27 |
| K-means[a] | 10.7 | 32.9 | 58.93 | 28.46 | 27.00 |
| K-means w/o grad norm (equation 7)[a] | 10.8 | 32.3 | 58.21 | 27.56 | 26.51 |

[a] Both use 256-d PCA for dimensionality reduction. Otherwise the gradient outliers result in multiple clusters with few data points.

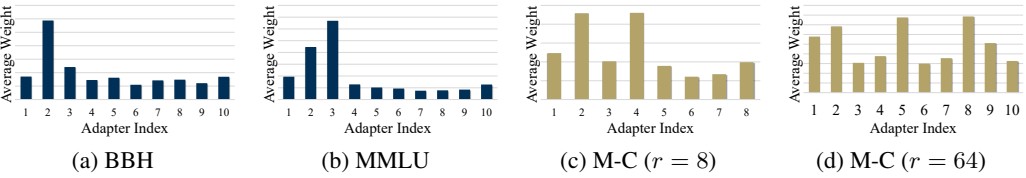

(a) BBH      (b) MMLU      (c) M-C ($r = 8$)      (d) M-C ($r = 64$)

Figure 2: Average weight distribution across clusters for different datasets and LoRA ranks. Only relative values matter. "M-C" represents MATH-Combined.

runs, especially at higher sampling temperatures. The Instruction Embedding baseline also falls short of ELREA, highlighting the critical role of a refined gradient profile in achieving optimal expertise extraction and routing.

When Gemma2-9b is used as the underlying architecture, ELREA still continues to outperform the base adapter, although with a narrower margin. The advanced capabilities of Gemma2-9b in capturing task-specific knowledge, even without explicit fine-tuning, appear to diminish the advantages of ELREA (Gemma Team, 2024a). It suggests that ELREA is more beneficial when the backbone model is less tailored to the task or when the fine-tuning dataset is more diverse and complex.

**Ablation Studies** An examination of Tables 1 and 2 shows that the gradient-based clustering method consistently outperforms the random approach. This underscores the effectiveness of gradient-based clustering in isolating in-domain, task-specific data for fine-tuning. However, the advantage of ELREA over the Random Cluster is not always prominent. This is understandable, considering that Random Cluster approximates Deep Ensembles, a very strong baseline that sufficiently exploits the training data. The inferior performance of the Uniform Weights baseline highlights the importance of a properly designed routing mechanism in ELREA. Figure 2 illustrates the average weight distribution across clusters. We observe that, for the in-domain MATH-Combined test set, the experts are more evenly activated across different data points. In contrast, the BBH and MMLU datasets exhibit a skewed distribution favoring one or two clusters with significantly higher average weights. In these latter cases, the test distribution accounts for only a small portion of the training data, likely dominated by a few clusters. This may also explain why the LESS method introduced by Xia et al. (2024) can outperform the baseline using fewer training data.

As noted by Xia et al. (2024), the dimensionality of the gradient projection $d_{\text{proj}}$ significantly influences the performance of training-test similarity matching. Figure 3a demonstrates a similar pattern for ELREA. When $d_{\text{proj}}$ is reduced from 8,192 to 512, there is a noticeable decline in ELREA' exact-match accuracy. This reduction compromises the model's ability to retain task-specific, fine-grained information, as random projection is more likely to omit essential features, resulting in diminished performance. Furthermore, an interesting observation on the BBH dataset reveals that ELREA underperforms compared to the base adapter at a projection dimensionality of 512, and even more so in comparison to the Random Cluster. Additionally, Table 3 shows that using PCA for dimensionality reduction, instead of selecting a smaller $d_{\text{proj}}$, also hurts performance. Similarly, using k-means for clustering degrades performance. This further underscores the importance of preserving representative gradient features for effective data clustering and matching, highlighting that failure to do so significantly impairs model performance.

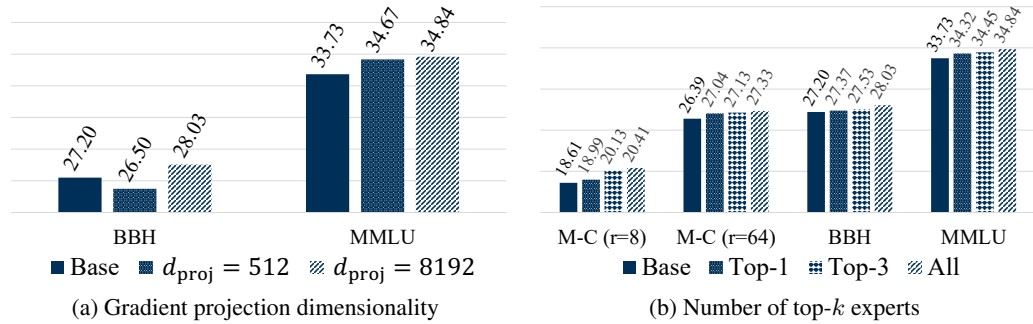

(a) Gradient projection dimensionality

(b) Number of top-$k$ experts

Figure 3: Effects of gradient projection dimensionality and selection of top-$k$ experts during inference on model performance.

Additionally, Figure 3b demonstrates that ELREA' performance improves with the number of top-$k$ experts selected during inference. This suggests that the model benefits from incorporating a diverse set of experts, even when the contribution of some experts is relatively minor. While selecting fewer experts can lead to more efficient inference, a trade-off must be carefully considered to balance performance with computational cost.

## 6 CONCLUSION

We introduced Ensembles of Low-Rank Expert Adapters (ELREA), a framework designed to address the challenge of conflicting gradient directions during the fine-tuning of LLMs across diverse datasets. ELREA develops multiple LoRA expert adapters, each optimized for a specific data cluster with similar gradient profiles. These adapters collaboratively generate predictions by dynamically adjusting their contributions based on the input's gradient characteristics, effectively resolving gradient conflicts without the need for task-specific data features or validation sets. Our approach, which requires only a single training session, enhances the adaptability of models to new or evolving tasks and outperforms traditional LoRA adapters and other ensemble techniques across a variety of applications. Ablation studies confirm that both the ensemble structure and the gradient-based clustering and routing mechanisms are integral to ELREA's effectiveness. These findings underscore the framework's potential for efficient and scalable application of LLMs in practical settings.

## LIMITATIONS

Compared to MoE approaches, ELREA incurs higher computational overhead during inference because multiple expert adapters are activated simultaneously. In our implementation, we replicate each input instance across the batch dimension and feed each copy to a distinct expert adapter. While this strategy reduces inference latency, it increases memory consumption, as shown in appendix G. Advanced adapter architectures, such as FLoRA (Wen & Chaudhuri, 2024), may help mitigate these computational demands by reducing matrix multiplication operations, which we leave for future work. Due to limited computational resources, our experiments focus on smaller-scale backbone LLMs and expert adapters. We observe that the performance gains of ELREA over the primary baseline, $\mathcal{M} + \mathcal{Q}_{\text{base}}$, diminish when the backbone LLM is already strong or well-adapted to the target task. Consequently, ELREA may be more beneficial when the backbone LLM is limited in capacity or when the target task significantly diverges from the pretraining materials. Our hyperparameter tuning for both ELREA and the baseline models is preliminary. A more thorough exploration of various clustering and routing methods could further enhance ELREA's performance and offer deeper insights into the model's behavior.

## ACKNOWLEDGMENTS

Supported by Amazon Development Center U.S., Inc. We thank Wei Ding, Baris Coskun, Zhilu Zhang, Chen Ling, Le Yu, Xuan Zhang, Xiayan Ji, and Zekuan Liu for helpful discussions.

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

## A    RELATED WORKS: MIXTURE OF EXPERTS AND DEEP ENSEMBLES FOR LANGUAGE MODELS

Mixture of Experts (MoE) have gained popularity in the field of LLM pre-training (Fedus et al., 2022; Jiang et al., 2024; Dai et al., 2024; Zhong et al., 2024) and fine-tuning (Gou et al., 2023; Shen et al., 2023; Luo et al., 2024; Zhou et al., 2024; Li et al., 2024a; Yang et al., 2024) as an approach to maintain model performance while reducing computational cost during inference. In LLM fine-tuning with MoE, the most frequent setup involves using each LoRA adapter, or simply a linear layer, as an expert, and employing a routing mechanism to select the most relevant adapters for each input token. The expert networks can be placed in parallel at different levels of the network modules (Cai et al., 2024), such as the feed-forward layers after multi-head attention (Dou et al., 2023; Diao et al., 2023; Li et al., 2024a), the linear layer within the attention block (Gou et al., 2023; Zhu et al., 2023; Luo et al., 2024; Tian et al., 2024), the Transformer block (Gao et al., 2024), or a combination of the above (Zadouri et al., 2024; Wu et al., 2024). In terms of routing, most works rely on trainable gating networks to predict the weights for each expert (Wang et al., 2023c; Li et al., 2024a; Wu et al., 2024; Chen et al., 2024; Liu et al., 2024a; Luo et al., 2024; Zadouri et al., 2024). Other studies leverage domain information or task-specific features to guide the routing process (Huang et al., 2024; Muqeeth et al., 2024; Liu et al., 2024b; Li et al., 2024a; Shen et al., 2024). Among these, the works most similar to ELREA are Gou et al. (2023), Zhou et al. (2024), and Yang et al. (2024), which use textual or gradient-based features to guide the routing process. Specifically, Gou et al. (2023) propose MoCLE, which first clusters the instruction embeddings using K-means and then trains a gating network to predict the top-$k$ cluster assignments for each token. Zhou et al. (2024) design a task-wise decorrelation loss to encourage the router to learn oriented weight combinations of adapters tailored to homogeneous tasks. Yang et al. (2024) route an input token to the expert that generates gradients not conflicting with the average gradient of the entire sequence.

Researchers have also explored the potential of applying Deep Ensembles to LLM pre-training and fine-tuning (Havasi et al., 2021; Tran et al., 2022; Cha et al., 2021; Liu et al., 2022; Gleave & Irving, 2022; Chronopoulou et al., 2023; Wang et al., 2023a; Jiang et al., 2023; Chen et al., 2023; Lu et al., 2024), as well as to related modules such as reward model learning (Coste et al., 2024; Zhang et al., 2024a; Ahmed et al., 2024) for reinforcement learning from human feedback (RLHF; Ouyang et al., 2022). Conventional Deep Ensemble methods train multiple models, or multiple LoRA adapters in the context of LLMs, on similarly distributed data and then average the predictions of these models to make the final prediction (Wang et al., 2023a; Coste et al., 2024). Another line, often referred to as "fusion", trains multiple models on heterogeneous data and then fuses the predictions of these models to make the final prediction (Jiang et al., 2023; Chen et al., 2023; Lu et al., 2024; Wang et al., 2024). Such works often do not impose any constraints on the model architectures, and the key to their success lies in how to select and combine the results from different models. For example, Jiang et al. (2023) propose a pairwise comparison method to effectively discern subtle differences between candidate outputs and enhance ranking performance for reward modeling. Wang et al. (2024) address the scenario of solving a task that requires different expertise scattered across multiple models and propose a fusion method based on $k$-nearest neighbors classifiers and a graph shortest path analogy to effectively combine the results of different models and achieve better performance.

## B    DATASETS

To evaluate the effectiveness of ELREA, we conducted experiments across two distinct categories: 1) general language understanding and reasoning, and 2) mathematical reasoning. Each category utilizes its own dedicated training and evaluation datasets, as detailed in Table 4.

**General Language Understanding and Reasoning**    For the first category, we followed the methodologies outlined in Xia et al. (2024) and Wang et al. (2023d). We employed a diverse combination of datasets for fine-tuning our model:

- **Flan V2** (Longpre et al., 2023): This comprehensive collection encompasses over 1,800 NLP tasks, combining numerous existing datasets with various data augmentations. The tasks cover a wide range of NLP problems, including question answering, summarization, translation, and sentiment analysis.

- **Chain-of-Thought (CoT)** (Wei et al., 2022; Longpre et al., 2023): A subset of the Flan V2 collection, the CoT dataset includes tasks annotated with chain-of-thought reasoning steps. It emphasizes the model's ability to generate intermediate reasoning processes, enhancing performance on complex tasks that require multi-step reasoning.
- **Dolly-15k** (Conover et al., 2023): This curated dataset contains approximately 15,000 high-quality, human-generated prompt-response pairs designed specifically for instruction tuning of LLMs. Created by Databricks employees, it focuses on instruction-following capabilities across a variety of domains and task types.
- **OpenAssistant Conversations** (Köpf et al., 2023): A multilingual, human-generated, and human-annotated assistant-style conversation corpus featuring fully annotated conversation trees in different languages. For our experiments, we utilize only the supervised fine-tuning portion of this dataset, excluding any content related to reward modeling or reinforcement learning.

These datasets vary significantly in size, format, tasks, and domains, providing a comprehensive training ground for general language understanding and reasoning. Specifically, Flan V2 and CoT datasets contribute to the model's ability to handle a wide range of NLP tasks with enhanced reasoning capabilities, while Dolly-15k and OpenAssistant Conversations improve the model's instruction-following and conversational skills. In practice, we directly use the pre-processed dataset provided by Xia et al. (2024), which consolidates these datasets into a unified format suitable for fine-tuning.[5]

For testing, we utilize two challenging benchmark datasets to evaluate the general reasoning and problem-solving abilities of our model:

- **Massive Multitask Language Understanding** (MMLU; Hendrycks et al., 2021a): MMLU is a comprehensive evaluation benchmark that assesses a model's knowledge and reasoning across 57 subjects, including humanities, sciences, social sciences, and more. The dataset consists of over 19,000 multiple-choice questions designed to mimic the difficulty of an average professional or college-level exam. Each question has four answer options, and the dataset provides only the correct answer without any accompanying reasoning or explanation.
- **BIG-Bench Hard** (BBH; bench authors, 2023; Suzgun et al., 2023): BBH is a subset of the BIG-Bench, consisting of 23 tasks identified as particularly challenging for LLMs. The tasks cover a diverse range of domains such as logical reasoning, mathematics, commonsense reasoning, *etc.*. Unlike MMLU, BBH includes not only the correct answers but also detailed CoT reasoning annotations for each question. This allows for the assessment of a model's ability to perform complex reasoning and generate intermediate reasoning steps.

Both datasets predominantly feature difficult multiple-choice question-answering formats with diverse question types, and only a few require numerical responses. The inclusion of reasoning chains in BBH enables a more in-depth evaluation of the model's reasoning capabilities compared to MMLU, which focuses solely on the final answers. Importantly, there is no significant overlap between the training datasets and these test datasets, ensuring that the evaluation measures the model's ability to generalize to unseen tasks and domains. To facilitate the desired output formatting and to guide the model during inference, we provide up to three in-context examples from the validation subset of the BBH dataset and five examples from MMLU dataset. These examples serve as prompts to help the model understand the expected answer format and improve its performance on the evaluation tasks.

**Mathematical Reasoning**   For the mathematical reasoning category, we developed the MATH-Combined dataset by integrating several existing mathematical problem-solving resources into a unified format analogous to the MATH dataset (Hendrycks et al., 2021b), including

- **GSM8K** (Cobbe et al., 2021): A dataset containing 8,000 high-quality grade school math word problems that require multi-step reasoning to solve. Each problem includes a question and a detailed step-by-step solution.
- **MathQA** (Amini et al., 2019): Originally a multiple-choice dataset derived from the AI2 Arithmetic and the DeepMind Mathematics datasets, MathQA consists of over 37,000 math word problems across various topics. Each problem comes with a question, multiple-choice answers, and annotated solution programs.

---

[5]Available at https://huggingface.co/datasets/princeton-nlp/less_data.

Table 4: Dataset statistics. Although listed separately here, the fine-tuning datasets are mixed together and randomly shuffled before being used for model fine-tuning or clustering.

| | Dataset | Source | # Instance | $l_{\text{instr}}$[a] | $l_{\text{resp}}$[a] |
|---|---|---|---|---|---|
| **General Language Understanding and Reasoning** | | | | | |
| Fine-Tune | Dolly-15k | Conover et al. (2023) | 15,011 | 72.41 | 60.12 |
| | OpenAssistant | Köpf et al. (2023) | 55,668 | 20.14 | 113.09 |
| | CoT | Wei et al. (2022) | 100,000 | 168.70 | 34.94 |
| | Flan V2 | Longpre et al. (2023) | 100,000 | 216.59 | 16.71 |
| Test | BBH | Suzgun et al. (2023) | 6,511 | 64.87[b] | 105.51 |
| | MMLU | Hendrycks et al. (2021a) | 14,042 | 88.53[b] | 1 |
| **Mathematical Reasoning (MATH-Combined)** | | | | | |
| Fine-Tune & Test | MATH | Hendrycks et al. (2021b) | 7,500 & 1,000 | 32.69 | 88.47 |
| | GSM8k | Cobbe et al. (2021) | 7,441 & 1,000 | 45.19 | 56.93 |
| | SVAMP | Patel et al. (2021) | 677 & 280 | 31.66 | 28.15 |
| | MathQA | Amini et al. (2019) | 26,287 & 998 | 38.39 | 69.09 |

[a] These numbers represent the average number of words (character strings separated by whitespace and newline characters) in the instruction and response sequences. They are generally smaller than the number of tokens.

[b] These numbers do not include the in-context examples; if the examples are considered, the counts will be approximately $3\times$ larger for BBH and $5\times$ larger for MMLU.

- **SVAMP** (Patel et al., 2021): A dataset designed to test the robustness of math word problem solvers by introducing subtle variations to existing problems. It contains 1,000 problems that require careful reasoning to avoid common pitfalls.
- **MATH** (Hendrycks et al., 2021b): A collection of 12,500 challenging competition-level math problems covering subjects like algebra, geometry, calculus, and more. Each problem includes a question and a detailed solution formatted in LaTeX.

To create a consistent and unified dataset, we process the inputs from GSM8K, MathQA, and SVAMP to match the format of the MATH dataset. We utilize Claude 3 Sonnet (Anthropic, 2024) to reformulate the final answers into a specified format, specifically using the "\boxed{}" command to enclose final answers. For MathQA, which is originally in a multiple-choice format, we retain only the correct answers and reformat them into value prediction tasks. This standardization ensures that all problems across the datasets have a uniform presentation, facilitating knowledge transfer and model training. During the processing, the reformatted outputs generated are compared to the original answers to ensure accuracy. If the model fail to produce the correct answer after five attempts, those instances are discarded to maintain the dataset quality.

Unlike the first category of general language understanding and reasoning, the fine-tuning and test datasets in MATH-Combined are similarly distributed. This alignment allows us to gain insights into the effectiveness of selecting task-specific data for fine-tuning, as it enables us to assess how well the model performs on tasks that closely resemble its training data. To manage computational resources efficiently, we sub-sample the test instances to approximately 1,000 problems per dataset. Preliminary experiments show that it provides a representative enough evaluation of the model's performance while reducing the computational burden.

## C  MODEL CONFIGURATIONS

Our primary experiments utilize Gemma-2b (Gemma Team, 2024b), which contains 2.5 billion network parameters, as the core framework for their relative efficiency in training and inference. Specifically, we employ the instruction-tuned variant gemma-1.1-2b-it, known for its efficiency in smaller-scale settings. We also conduct experiments with the larger and more advanced Gemma2 model gemma-2-9b-it (Gemma Team, 2024a) to investigate the impact of backbone model representativeness on the relative performance.[6] For the LoRA modifications, we default to a rank

---

[6]Available at https://huggingface.co/google/gemma-2-9b-it.

$r = 8$ across all linear layers in the model (*i.e.*, {q_proj, k_proj, v_proj, o_proj, up_proj, down_proj, gate_proj}), which count as about $0.39\%$ of the total network parameters. In a separate experiment targeting the MATH-Combined dataset, we also explore the impact of increasing the rank to $r = 64$. The adapter's scaling factor $\alpha$ and dropout rate are consistently set to $\alpha = 4r$ and $p_{\text{dropout}} = 0.1$, respectively. The architecture for cluster-wise adapters $\mathcal{Q}_c$ mirrors that of the base adapter $\mathcal{Q}_{\text{base}}$ to streamline implementation. We typically set the gradient projection dimensionality to $d_{\text{proj}} = 8192$, but also include experiments with $d_{\text{proj}} = 512$ to investigate the impact of dimensionality reduction on model performance.

Due to license restrictions, we are unable to use LLaMA-series models (Touvron et al., 2023) for our experiments.

## D    FINE-TUNING

For both dataset categories, we fine-tune the base adapter $\mathcal{Q}_{\text{base}}$ for 2 epochs using the Adam optimizer, with an initial learning rate of $5 \times 10^{-5}$ that linearly decays to zero. Preliminary testing indicates that 2 epochs optimize performance for $\mathcal{Q}_{\text{base}}$, ensuring a fair comparison with our method. We also observe a strong tendency toward overfitting beyond this point, as indicated by the loss value and gradient norm curve. Cluster-wise adapters $\mathcal{Q}_c$ undergo an identical duration of fine-tuning at a slightly reduced learning rate of $2 \times 10^{-5}$. These hyperparameters, derived from prior experience, are fixed without adjustments to preemptively accommodate unseen test data, diverging from the methods of Xia et al. (2024). Most fine-tuning sessions are conducted on an computing instance equipped with 8 NVIDIA A100 40GB GPUs, employing 4-bit quantization for the backbone model $\mathcal{M}$ and bf16 precision for adapters $\mathcal{Q}$. This setup essentially uses QLoRA (Dettmers et al., 2023) rather than LoRA, but we do not specifically distinguish them as they both belong to the LoRA family and do not impact our conclusions. Additional training sessions utilize instances with 8 NVIDIA V100 32GB GPUs, using fp16 precision. We observe no difference in performance between these configurations apart from training speed. The maximum token sequence length for training is 2,048, with a batch size of 16 sequences distributed across the GPU instances. Only a few ($< 100$ for each dataset category using the Gemma-2b tokenizer) of training sequences are longer than this threshold, and we simply discard these instances.

## E    BASELINES

Our primary baseline is the **base LoRA adapter** $\mathcal{M} + \mathcal{Q}_{\textbf{base}}$, which is fine-tuned on the complete dataset for 2 epochs to achieve optimal performance, as detailed in Section D. Additionally, we consider a **dataset-wise adapter** $\mathcal{M} + \mathcal{Q}_{\textbf{dataset}}$ for MATH-Combined, where the adapter is fine-tuned and applied to each test subset individually. For instance, $\mathcal{M} + \mathcal{Q}_{\text{MATH}}$ is fine-tuned on the MATH training subset of MATH-Combined and evaluated on its corresponding MATH test subsets; similarly, $\mathcal{M} + \mathcal{Q}_{\text{GSM8K}}$ is fine-tuned on the GSM8K training subset and evaluated on the GSM8K test subsets, and so on. dd We also include the **backbone model** $\mathcal{M}$ itself as a baseline, which is used directly for test-case inference without any adapter fine-tuning. This baseline is applied only to BBH and MMLU datasets, as they contain in-context examples to guide the model's output format. All other baseline methods start from the $\mathcal{M} + \mathcal{Q}_{\text{base}}$ checkpoint for further fine-tuning or inference, and include:

- **MoE Routing**: This baseline implements layer-level routing with the same weights as EL-REA. Specifically, similar to equation 3, the averaged linear layer adapter output is given by

$$\mathcal{F}(\boldsymbol{x}) = \sum_{c=0}^{C} \lambda_c \boldsymbol{B}_c \boldsymbol{A}_c^{\mathsf{T}} \boldsymbol{x}; \quad \lambda_c = \frac{w_c}{\sum_{c'=0}^{C} w_{c'}}; \quad w_0 \triangleq w_{\text{base}}, \boldsymbol{A}_0 \triangleq \boldsymbol{A}_{\text{base}}, \boldsymbol{B}_0 \triangleq \boldsymbol{B}_{\text{base}}. \quad (11)$$

  Here, we omit the layer indicator $i$ for simplicity. The matrices $\boldsymbol{A}$ and $\boldsymbol{B}$ are defined as in § 2.1, and $w$ represents the routing weight for each cluster as in equation 9. Note that $\mathcal{F}(\boldsymbol{x})$ is the output of the LoRA MoE, which should be added to the layer output from the backbone model $\mathcal{M}(\boldsymbol{x})$ with a scaling factor of $\alpha/r = 4$, as mentioned in Appendix C.
- **MoE Merging**: This baseline merges the expert network weights before processing the input. Specifically, the averaged linear layer adapter weights become the final weights for the model,

*i.e.*, $\boldsymbol{A} = \sum_{c=0}^{C} \lambda_c \boldsymbol{A}_c$ and $\boldsymbol{B} = \sum_{c=0}^{C} \lambda_c \boldsymbol{B}_c$. Once merged, the network behaves as a single-expert model, and the output is calculated as $\mathcal{F}(\boldsymbol{x}) = \boldsymbol{B}\boldsymbol{A}^\mathsf{T}\boldsymbol{x}$.

- **Mixture of LoRA Experts** (**MoLE**, Wu et al., 2024): This baseline models each layer of trained LoRAs as a distinct expert and incorporates a learnable gating function within each layer, in contrast to the precomputed universal routing weights used in MoE Routing. Using the same notation as in equation 11, the output of each MoLE layer is defined as

$$\mathcal{F}(\boldsymbol{x}) = \sum_{c=0}^{C} \lambda_c \boldsymbol{B}_c \boldsymbol{A}_c^\mathsf{T} \boldsymbol{x}; \quad \lambda_c = \frac{\exp(\boldsymbol{w}_c^\mathsf{T}\boldsymbol{x})}{\sum_{c'=0}^{C} \exp(\boldsymbol{w}_{c'}^\mathsf{T}\boldsymbol{x})}, \tag{12}$$

where $\boldsymbol{w}_c$, a vector of the same dimensionality as $\boldsymbol{x}$, represents the learnable gating weight of a single-output linear layer for each expert $c$. In our setup, the gating outputs are expected to exhibit an imbalanced distribution, as shown in Figure 2. Consequently, we do not include the gating balancing loss proposed by Wu et al. (2024). The routing parameters are trained on the entire training set for 1 epoch at a learning rate of $2 \times 10^{-5}$ with all other parameters frozen.

- **LoRA Ensembles** (Wang et al., 2023a): This baseline trains three adapters, $\mathcal{Q}_1$, $\mathcal{Q}_2$, and $\mathcal{Q}_3$, independently on the entire dataset using the same configuration as the base adapter $\mathcal{Q}_{\text{base}}$ (§ 3.1). During inference, four models (*i.e.*, $\{\mathcal{M} + \mathcal{Q}_{\text{base}}^{(e)}\}$ and $\{\mathcal{M} + \mathcal{Q}_i\}_{i=1}^{3}$) are applied to the input sequence. The final prediction is then computed by averaging their pre-activation logits and taking the ArgMax as the predicted next token. We do not match the number of ensemble models to the number of clusters, $C$, in ELREA due to concerns about the training and evaluation costs.

- **Self-Consistency** (Wang et al., 2023b): This baseline performs 5 separate inference passes with $\mathcal{M} + \mathcal{Q}_{\text{base}}$ for each instance, using random token sampling with the last-layer SoftMax activation temperature set to 1. The final answer is determined by majority voting among the 5 predictions. In case of a tie, one of the tied answers is randomly selected as the final prediction.

- **Instruction Embedding**: Instead of using the instruction gradients representation from equation 2, this baseline employs the sentence embedding of the instruction text directly for training data clustering and test instance routing. Specifically, we use the Sentence Transformers (Reimers & Gurevych, 2019) Python package with the `all-mpnet-base-v2` model checkpoint[7] to encode the instruction text into a fixed-size vector, which is then used for clustering and routing in the same way as the gradient features.

- **Random Cluster**: This baseline maintains the same number of clusters and cluster sizes as ELREA but assigns cluster members randomly from the fine-tuning dataset $\mathcal{D}_{\text{ft}}$. Specifically, $\mathcal{D}_{\text{rand},c} \subset \mathcal{D}_{\text{ft}}$, with $|\mathcal{D}_{\text{rand},c}| = |\mathcal{D}_c|$, and $\mathcal{D}_{\text{rand},c} \cap \mathcal{D}_{\text{rand},c'} = \varnothing$ for all $c \neq c' \in \{1, 2, \ldots, C\}$. The corresponding adapters are fine-tuned on these randomly assigned clusters and are uniformly weighted during inference, *i.e.*, $w_{\text{rand,base}} = w_{\text{rand},1} = \ldots = w_{\text{rand},C} = 1$. This random assignment preserves the distribution characteristics of $\mathcal{D}_{\text{ft}}$, positioning Random Cluster as an approximate deep ensemble baseline with equivalent training effort to ELREA.

- **Uniform Weights**: This baseline assigns uniform weights to all clusters during inference, *i.e.*, $w_{\text{base}} = w_1 = \ldots = w_C = 1$.

## F INFERENCE PROMPTING

Listing 1: An example of MATH-Combined inference prompts.

```
1 <bos><start_of_turn>user
2 Let $A = (2, 0)$, $B = (0, 2)$, $C = (-2, 0)$, and $D = (0, -2)$.  Compute the greatest
      possible value of the product $PA \cdot PB \cdot PC \cdot PD$, where $P$ is a point on
      the circle $x^2 + y^2 = 9$.<end_of_turn>
3 <start_of_turn>model
```

Listing 2: An example of expected model answer for dataset MATH-Combined.

```
1 We use complex numbers. Let $a = 2$, $b = 2i$, $c = -2$, and $d = -2$ be the complex numbers
      corresponding to $A$, $B$, $C$, and $D$, respectively. Let $p$ be the complex number
      corresponding to $P$, so that $|p| = \sqrt{9} = 3$. Then we have \[\begin{aligned} PA \
      cdot PB \cdot PC \cdot PD &= |p-2| \cdot |p-2i| \cdot |p+2| \cdot |p+2i| \\ &= |(p-2)(p
      +2)| \cdot |(p-2i)(p+2i)| \\ &= |p^2-4| \cdot |p^2+4| \\ &= |p^4-16|. \end{aligned}\]
      Since $|p| = 3$, we have $|p^4| = 3^4= 81$, so by the triangle inequality, \[|p^4-16| \le
```

---

[7] https://huggingface.co/sentence-transformers/all-mpnet-base-v2.

```
    |pˆ4| + |-16| = 81 + 16 = 97.\]Equality holds if and only if $pˆ4 = -81$, which occurs
    when $p = 3\left(\frac{\sqrt2}{2} + \frac{\sqrt2}{2}i\right)$. Therefore, the answer is $
    \boxed{97}$.<end_of_turn>
2 <eos>
```

Listing 3: An example of BBH inference prompts.

```
1 <bos><start_of_turn>user
2 Infer the date from context.
3
4 Example 1:
5 Q: Today is Christmas Eve of 1937. What is the date 10 days ago in MM/DD/YYYY?
6 Options:
7 (A) 12/14/2026
8 (B) 12/14/1950
9 (C) 12/14/2007
10 (D) 12/14/1937
11 (E) 07/14/1938
12 (F) 12/14/1988
13 A: Let's think step by step.
14 If today is Christmas Eve of 1937, then today's date is December 24, 1937. 10 days before
       today is December 14, 1937, that is 12/14/1937. So the answer is (D).
15
16 Example 2:
17 Q: Tomorrow is 11/12/2019. What is the date one year ago from today in MM/DD/YYYY?
18 Options:
19 (A) 09/04/2018
20 (B) 11/11/2018
21 (C) 08/25/2018
22 (D) 11/02/2018
23 (E) 11/04/2018
24 A: Let's think step by step.
25 If tomorrow is 11/12/2019, then today is 11/11/2019. The date one year ago from today is
       11/11/2018. So the answer is (B).
26
27 Example 3:
28 Q: Jane and John married on Jan 2, 1958. It is their 5-year anniversary today. What is the
       date tomorrow in MM/DD/YYYY?
29 Options:
30 (A) 01/11/1961
31 (B) 01/03/1963
32 (C) 01/18/1961
33 (D) 10/14/1960
34 (E) 01/03/1982
35 (F) 12/03/1960
36 A: Let's think step by step.
37 If Jane and John married on Jan 2, 1958, then and if it is their 5-year anniversary today,
       then today's date is Jan 2, 1963. The date tomorrow is Jan 3, 1963, that is 01/03/1963.
       So the answer is (B).
38
39 Question:
40 Q: Today is Christmas Eve of 1937. What is the date tomorrow in MM/DD/YYYY?
41 Options:
42 (A) 12/11/1937
43 (B) 12/25/1937
44 (C) 01/04/1938
45 (D) 12/04/1937
46 (E) 12/25/2006
47 (F) 07/25/1937<end_of_turn>
48 <start_of_turn>model
```

Listing 4: An example of MMLU inference prompts.

```
1 <bos><start_of_turn>user
2 Please solve the following multi-choice problems.
3
4 Example 1:
5 What distinguishes coercive diplomacy from military force?
6
7 Option A: Compellence is another term for coercive diplomacy, but covering a narrower set of
       criteria; compellence covers those threats aimed at initiating adversary action. A threat
        to coerce a state to give up part of its territory would count as coercive diplomacy, as
        long as that threat proactively initiates action before reactive diplomacy is taken.
8 Option B: Coercive diplomacy constitutes the threats of limited force to induce adversary's
       incentive to comply with the coercer's demands. It is an influence strategy that is
       intended to obtain compliance: the use of force to defeat an opponent first does not
       count. It leaves an element of choice with the target to comply, or to continue.
9 Option C: Military force, or the threat of military force, utilises fear to achieve strategic
       objectives. Coercive diplomacy is differentiated from this approach, because it does not
       use fear as a tool for coercing an adversary.
```

```
10 Option D: Coercive diplomacy is employed to use force but to limit its effects on the
        international community. Coercive diplomacy is an aggressive strategy that is intended to
         obtain compliance through defeat. It does not leave an element of choice with the target
        , the target either being forced to comply or engage in conflict. It seeks to control by
         imposing compliance by removing any opportunity for negotiation or concession.
11
12 Answer: B
13
14 Example 2:
15 Which of the following is the best lens through which to investigate the role of child
        soldiers?
16
17 Option A: Child soldiers are victims of combat that need re-education and rehabilitation.
18 Option B: Children and their mothers are not active subjects in warfare and are best
        considered as subjects in the private sphere.
19 Option C: Children are most often innocent bystanders in war and are best used as signifiers
        of peace.
20 Option D: Children have political subjecthood that is missed when they are considered as
        passive victims of warfare.
21
22 Answer: D
23
24 Example 3:
25 In order to become securitized, a threat must be presented in which of these ways?
26
27 Option A: As an existential threat that requires immediate and extraordinary action, posing a
        threat to the survival of the state or to societal security.
28 Option B: As requiring immediate and extraordinary action by the state, threatening the
        survival of a referent object and therefore warranting the use of measures not normally
        employed in the political realm.
29 Option C: As an urgent threat to the survival of the referent object, so serious that it
        legitimises the employment of extraordinary action in response.
30 Option D: As an urgent threat to the survival of the audience that requires extraordinary or
        emergency measures.
31
32 Answer: C
33
34 Example 4:
35 How can we best describe the relationship between the state-centric approach and the concept
        of human security?
36
37 Option A: There are such wide divisions within the human security framework regarding the
        nature of threats and referent objects that no widely applicable comparisons between
        state-centric approaches and human security can be drawn.
38 Option B: By adopting the framework of human security, the limitations of the realist state-
        centric approach become evident. Whilst human security defines the referent object as the
         person or population, state-centric approaches prioritise the security of the state, de-
        prioritizing the pursuit of human security.
39 Option C: The state-centric approach to security is a faction of human security, usually
        defined within the broad school of human security. By being state-centric this approach
        prioritises the individual as the referent object in security studies.
40 Option D: Both the state-centric and human-centric approaches to security are mutually
        exclusive and offer a sufficient analytic framework with which to understand the
        international security system. It is therefore the role of security analysts to determine
         which of these substantial concepts is correct, and which should be discarded.
41
42 Answer: B
43
44 Example 5:
45 What are the frameworks of analysis within which terrorism has been considered (as of 2020)?
46
47 Option A: Competition between larger nations has resulted in some countries actively
        supporting terrorist groups to undermine the strength of rival states. Terrorist networks
         are extended patronage clubs maintained and paid for by their donor states and are
        conceptualised as being like state actors, to be dealt with using military force.
48 Option B: Globalization has enabled the internationalization of terrorist activities by
        opening up their operational space, although coordination is still managed from a
        geographical base. This suggests that terrorist groups are nationally structured which
        means that terrorism cannot be considered in terms of a war to be defeated militarily
        without having serious implications on the indigenous population.
49 Option C: Terrorism can be viewed as a problem to be resolved by military means (war on
        terrorism), by normal police techniques (terrorism as crime), or as a medical problem
        with underlying causes and symptoms (terrorism as disease).
50 Option D: Terrorism is viewed as a criminal problem. The criminalization of terrorism has two
        important implications. Firstly, it suggests that terrorism can be eradicated -
        terrorists can be caught and brought to trial by normal judicial proceedings thereby
        removing the threat from society - and secondly, it suggests that preventative crime
        techniques are applicable to prevent its development.
51
52 Answer: C
53
```

Table 5: Efficiency comparison on a toy dataset. Time is in seconds; memory is in GiB.

| Step | $\mathcal{M} + \mathcal{Q}_{\text{base}}$ | | ELREA | |
|---|---|---|---|---|
| | Time | Memory | Time | Memory |
| Fine-tuning base adapter $\mathcal{Q}_{\text{base}}$ on $\mathcal{D}_{\text{ft}}$ (§ 3.1) | 246 | 15.49 | 246 | 15.49 |
| Calculating training gradient features $\boldsymbol{\delta}(\boldsymbol{x}_{\text{ft, instr}})$ (§ 3.3) | – | – | 68 | 24.76 |
| Calculating test gradient features $\boldsymbol{\delta}_{\text{test}}$ (§ 3.4) | – | – | 14 | 24.76 |
| Fine-tuning experts on clusters (§ 3.3) | – | – | 246 | 15.49 |
| Fine-Tuning Total | 246 | – | 574 | – |
| Inference (§ 3.4) | 114 | 7.73 | 262 | 18.46 |

```
54  Question:
55
56  Which of these principles is not an element of the responsibility to protect?
57
58  Option A: The responsibility to prevent.
59  Option B: The responsibility to react.
60  Option C: The responsibility to remain sovereign.
61  Option D: The responsibility to rebuild.<end_of_turn>
62  <start_of_turn>model
```

## G  EFFICIENCY ANALYSIS

**Theoritical Analysis**  Theoretically, the computational overhead of ELREA compared to using $\mathcal{M} + \mathcal{Q}_{\text{base}}$ arises from the following aspects:

1) the computation of the gradients of all training and test instructions; 2) clustering the gradient features of the training data points and computing the weights of each test data point on the clusters; 3) additional training steps to fit LoRA experts on the training clusters; 4) additional computational resources required to perform the forward pass on all LoRA experts for each test data point. In practice, step 2) only takes a few minutes with our clustering setup (§ 3.3 and § 3.4), which is negligible compared to the entire training process and will be ignored in the following discussion.

If implemented properly, step 1) can also be integrated into the training and inference process with relatively small overhead. With a naïve implementation, step 1) approximately equals the cost of training the model on the combination of training and test *instructions* (without answers) for one epoch, whose overhead depends on the average length of the instructions. For datasets such as OpenAssistant, MATH, GSM8k, and MathQA, whose average instruction length is comparatively much shorter than the answer length (Table 4), the overhead is minimal. In the worst-case scenario, step 1)'s overhead approximates the cost of training the model on the combination of training and test for one epoch, which is still acceptable for most fine-tuning datasets.

As the sum of our training cluster sizes equals the number of training data points, *i.e.*, $\sum_{c=1}^{C} |\mathcal{D}_c| = |\mathcal{D}_{\text{ft}}|$, the additional training steps in step 3) take the same amount of time as training the base adapter $\mathcal{Q}_{\text{base}}$ (§ 3.4) on $\mathcal{D}_{\text{ft}}$, excluding CPU-disk I/O overhead, which is generally less than one minute in our experiments.

The complexity of step 4), however, is harder to estimate as it varies drastically according to the implementation. In our implementation, we choose to duplicate the input instruction along the batch dimension by the number of experts (*i.e.*, $C + 1$) and perform a forward pass on the backbone and all experts simultaneously. This implementation has a similar cost to using a $(C + 1)\times$ inference batch size with the base adapter $\mathcal{M} + \mathcal{Q}_{\text{base}}$.

**Empirical Results**  To evaluate the efficiency of ELREA, we compared its computation time with that of the baseline model $\mathcal{M} + \mathcal{Q}_{\text{base}}$ using a same set of hyper-parameters and device configuration on a single NVIDIA A101 80G GPU, except for the following specific parameters. We generate a **toy** dataset consisting of 2,000 training samples and 400 test samples as a smaller-scale but more controllable evaluation setup. Each sample contains 60 random *lorem-ipsum* words in both the instruction and the answer (which accounts for around 200 tokens each), matching the lengths in Dolly-15k (Table 4). We designate $C = 4$ experts and set the LoRA ranks to $r = 8$. The model

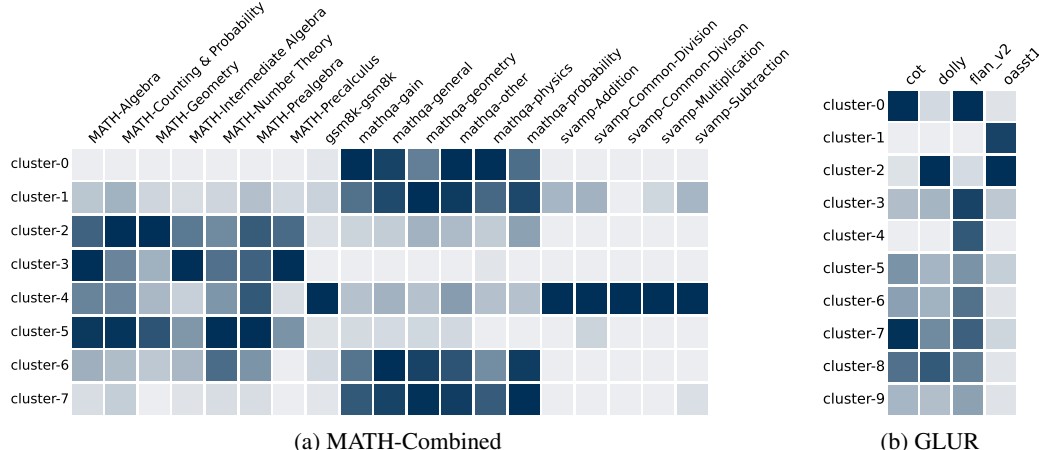

(a) MATH-Combined       (b) GLUR

Figure 4: Distribution of data sources and categories within each cluster for the MATH-Combined and GLUR (general language understanding and reasoning) training sets at rank $r = 8$. Cluster indices are shown along the rows, while columns represent data sources and categories, formatted as "{source dataset}-{category}" for MATH-Combined and "{source dataset}" for GLUR. The color intensity reflects the sample count, with darker shades indicating higher counts. Each column is independently normalized, meaning scales may differ across columns. Color gradients are slightly curved to improve visibility for categories with fewer samples.

undergoes fine-tuning over 3 epochs, with batch sizes of 4 for both fine-tuning and inference. During inference, the model consistently predict the next 20 tokens for all input instructions to ensure a fair comparison.

The results from our implementation, presented in Table 5, indicate that the fine-tuning time for EL-REA was 574 seconds, which is approximately $2.3\times$ that of the baseline $\mathcal{M} + \mathcal{Q}_{\text{base}}$'s 246 seconds. Similarly, the inference time and memory consumption are about $2.3\times$ and $2.4\times$, respectively. In contrast, a classic Deep Ensembles setup, where each LoRA expert is trained independently from scratch on the entire dataset, would require $5\times$ the time of the baseline for both fine-tuning and inference. Thus, ELREA offers significant efficiency and performance gains compared to this more traditional approach.

Further enhancements to ELREA' efficiency could be achieved by reducing the number of experts or the LoRA ranks, or by constructing gradient features from only the top-$k$ Transformer blocks rather than the entire model. Moreover, we are exploring LoRA merging techniques in ongoing work to effectively combine similar expert adapters, thereby further reducing inference costs.

## H    FURTHER ANALYSIS ON DATA CLUSTERING

To better understand the distribution of data across clusters, we analyzed the sources and categories within each cluster from the MATH-Combined dataset, as visualized in Figure 4. Here, "data source" refers to the individual datasets that comprise MATH-Combined (*i.e.*, MATH, GSM8k, SVAMP, or MathQA) and language understanding and reasoning (*i.e.*, CoT, Dolly-15k, Flan V2, and OpenAssistant), and "category" pertains to the finer-grain labels within these datasets. Notably, GSM8k is categorized uniformly under a single label "gsm8k" due to its lack of distinct category labels.

Analysis of Figure 4 reveals distinct correlations between clusters and data sources. For instance, in MATH-Combined, clusters 2, 3, and 5 predominantly contain samples from MATH, whereas clusters 0, 1, 6, and 7 primarily feature contributions from MathQA. This clustering also appears to group together tasks requiring similar mathematical skills; for example, cluster 4 heavily includes SVAMP samples, which typically assess algebraic problem-solving capabilities, alongside significant portions of "Algebra" and "Prealgebra" from the MATH dataset.

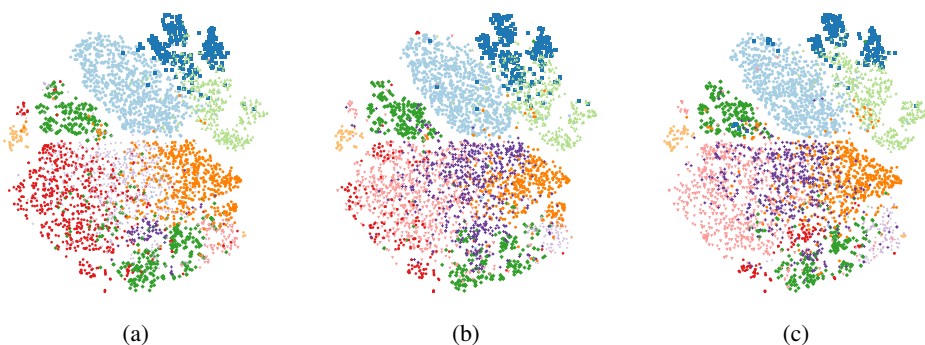

|     |     |     |
| --- | --- | --- |
| (a) | (b) | (c) |

Figure 5: Examples of data clusters from MATH-Combined, generated using different random seeds in cases where the clusters are non-identical. The entire dataset is used for clustering, but only $10\%$ of the data is visualized for clarity. The $8{,}192$-dimensional gradient features are projected into 2D space using t-SNE. The colors are randomly assigned; the same color does not necessarily imply the same cluster across different seeds.

Additionally, within individual sources, clusters distinguish between finer categories effectively; cluster 2 mainly focuses on Geometry and Probability, whereas cluster 3 is concentrated on Algebra. These insights suggest that the data representations successfully capture inherent structural differences, making the clustering both interpretable and meaningful. Such characteristics motivates the design of ELREA and significantly improves its efficacy.

As mentioned in § 3.3, the clustering process is robust to random seeds; *i.e.*, different seeds yield similar clusters. In cases where the clusters are not identical, we visualize them using t-SNE in Figure 5, which demonstrates sensible data partitioning and similar cluster structures across different seeds. Even if the cluster boundaries are not identical, the ensemble framework in ELREA effectively mitigates these differences through weighted aggregation of experts, ensuring robust performance across various cluster configurations. Therefore, the clustering process is both stable and reliable, providing a strong foundation for the ELREA framework.

