# OpenReview forum: "Ensembles of Low-Rank Expert Adapters"
_ICLR.cc/2025/Conference — ICLR 2025 Poster_

### Official Review · Reviewer_5M8L · 2024-10-29

**Soundness:** 2
**Presentation:** 3
**Contribution:** 2
**Rating:** 6
**Confidence:** 4

**Summary:**

This paper introduces Ensembles of Low-Rank Expert Adapters (ELREA), an ensemble-style LoRA MoE model to improve task adaptability and mitigate optimization conflicts in LLM fine-tuning. ELREA tackles gradient conflicts from diverse data sources by grouping tasks with similar gradient directions, creating specialized experts for each group. During inference, ELREA dynamically assigns weights to these experts based on gradient similarity with the input task instruction, selecting the most relevant experts for each task. Compared to traditional deep ensembles and standard MoE models, ELREA strikes a balance between performance and computational efficiency. Experimental results validate ELREA’s effectiveness.

**Strengths:**

- The paper is well-written and easy to follow.
- The model introduces a novel approach by designing experts based on gradient directions, rather than relying on an explicitly learned gating function, which is innovative in MoE design.
- The experiments demonstrate the model’s competitive performance across diverse tasks.

**Weaknesses:**

Overall, as the authors acknowledge in the limitations, the main weakness lies in efficiency and computational cost:

- In the inference stage, ELREA requires combining predictions from all expert adapters and the base adapter, resulting in inference costs that increase linearly with the number of experts, which can restrict practical applications.
- During inference, the model needs to first compute and store gradients for different task instructions.
- Additionally, clustering gradients during training introduces extra computational overhead.
- Although ELREA shows stronger performance compared to baselines like "MoE Routing", this advantage is somewhat diminished by its high inference costs.

The baseline implementations could be more complete:

- The current implementation of "MoE Routing" bypasses training the MoE gate and instead directly uses weights similar to ELREA’s ensemble approach. However, training the MoE gate separately at each layer would make the baseline comparison more robust.

The marginal benefit of ELREA diminishes as the backbone model scales while the inference cost significantly increases. Additionally, there is limited comparison with a 9B model in the experimental results.

**Questions:**

1. Could ELREA maintain most of its performance by the ensemble of sparse experts, like the top-4 or top-1 experts in addition to the base adapter, to reduce inference costs?

2. Regarding the completeness of baselines, there could be an additional version of "MoE Routing" similar to those implemented in MoLE [1] or LLaVA-MoLE [2]. Further comparisons with other LoRA MoE methods like LoraHub [3] will make the baseline comparisons more robust.

3. Is the clustering process a time-intensive part of the training process?

[1] Wu, Xun, Shaohan Huang, and Furu Wei. "Mixture of LoRA Experts." The Twelfth International Conference on Learning Representations, 2024, https://openreview.net/forum?id=uWvKBCYh4S.

[2] Chen, Shaoxiang, Zequn Jie, and Lin Ma. "Llava-mole: Sparse mixture of lora experts for mitigating data conflicts in instruction finetuning mllms." *arXiv preprint arXiv:2401.16160* (2024).

[3] Huang, Chengsong, et al. "Lorahub: Efficient cross-task generalization via dynamic lora composition." *arXiv preprint arXiv:2307.13269* (2023).

---

> ### Author Response · Authors · 2024-11-21
>
> Dear Reviewers,
>
> Thank you very much for your thoughtful review and valuable suggestions on our manuscript. We greatly appreciate your time and effort in evaluating our work. We have carefully considered your comments and have made corresponding revisions to address your concerns.
>
> ### Weaknesses
>
> 1. **Method efficiency**: We understand your concern regarding the efficiency and computational cost of our model. In response, we have conducted both theoretical and empirical analyses, which are now included in Appendix G "Efficiency Analysis" of the revised manuscript. Specifically, for a 4-expert system (in addition to the base adapter), we found that the fine-tuning and inference resource consumption is approximately 2.3 times that of using the base adapter alone. While this increase is not negligible, we believe it is acceptable in practice given the performance improvements. Nonetheless, we acknowledge that efficiency is crucial, and we are actively working on methods to reduce computational costs in our ongoing research.
>
> 2. **Baselines**: We appreciate your suggestion to include additional baselines for a more robust comparison. In the revised Section 5 "Results and Discussion," we have incorporated results from MoLE [1] and provided an in-depth discussion of its performance relative to our model. Regarding LoraHub [3], we found that its primary objective differs from ours, as it focuses on selecting and combining adapters for downstream unseen tasks based on a few examples. Since our work assumes in-domain test samples without task-specific examples during inference, we believe LoraHub may not be a directly comparable baseline. However, we acknowledge its relevance and have included a discussion of LoraHub in the related works section.
>
> ### Questions
>
> 1. **Top-k experts during inference**: Thank you for this insightful suggestion. Activating only the top-1 or top-4 experts during inference could indeed reduce computational costs, but it also leads to reduced performance. We illustrated these findings and provided a discussion in Figure 3(b) and the paragraph below (Lines 502--506) in the updated manuscript.
>
> 3. **Clustering efficiency**: We agree that clustering efficiency is an important consideration. Our clustering pipeline, as introduced in Section 3.3 "Clustering and Per-Cluster Fine-Tuning," is designed to be highly efficient. As mentioned in Appendix G "Efficiency Analysis," the clustering step requires minimal computational effort and does not significantly impact the overall training time.
>
> We hope that our revisions and additional analyses have addressed your concerns. Your feedback has been invaluable in improving the quality of our work. We hope that our revised manuscript meets the ICLR quality standards and would be grateful if you would consider updating your evaluation accordingly.
>
> Please do not hesitate to let us know if you have any further questions or suggestions. We are more than happy to engage in further discussions to enhance our work.
>
> Best regards,
> The Authors
>
> [1] Wu, Xun, Shaohan Huang, and Furu Wei. "Mixture of LoRA Experts." The Twelfth International Conference on Learning Representations, 2024, https://openreview.net/forum?id=uWvKBCYh4S.
> [2] Chen, Shaoxiang, Zequn Jie, and Lin Ma. "Llava-mole: Sparse mixture of lora experts for mitigating data conflicts in instruction finetuning mllms." arXiv preprint arXiv:2401.16160 (2024).
> [3] Huang, Chengsong, et al. "Lorahub: Efficient cross-task generalization via dynamic lora composition." arXiv preprint arXiv:2307.13269 (2023).

---

> > ### Comment · Reviewer_5M8L · 2024-11-22
> >
> > Thanks for the authors' responses, which have addressed my concerns. I raised my rating accordingly.

---

> ### Author Response · Authors · 2024-11-22
> **Thank you for your consideration**
>
> Dear Reviewer,
>
> Thank you again for taking the time to review our responses and for your thoughtful consideration. We are glad that we were able to address your concerns, and we appreciate you raising your rating.
>
> Best regards,
> The Authors

---

### Official Review · Reviewer_Ue2L · 2024-10-31

**Soundness:** 3
**Presentation:** 3
**Contribution:** 2
**Rating:** 6
**Confidence:** 4

**Summary:**

The paper explores the issue of gradient conflict during multi-source data training. It addresses this problem by clustering gradients and training LoRA for each cluster, followed by ensembling to avoid the conflicts. Subsequent experiments on multiple datasets validate the effectiveness of the proposed method.

**Strengths:**

Using gradient features for clustering provides a more direct solution to the gradient conflict issue compared to clustering based on sentence embeddings, resulting in improved performance.

**Weaknesses:**

1. The paper does not sufficiently justify the choice of ensembling over a mixture of experts (MoE) approach. It mentions that ensembling performs better in value prediction and uncertainty estimation; however, this may not hold true for NLP tasks. In fact, MoE can optimize routing at a finer granularity, which may yield better performance compared to ensembling. This should be discussed in the paper. Additionally, a comparison should be made during the second stage of training cluster-specific LoRA, where a router could be trained alongside (with similar computational overhead) instead of using ELREA's weights for routing.
2. The paper needs to address the algorithm's time complexity. The overall process is lengthy and involves additional gradient calculations, leading to higher computational costs compared to other methods. Discussing these costs is essential for the practical application of the proposed method.

**Questions:**

1. Since the training of adapters consists of two stages (base LoRA and cluster LoRA training), my understanding is that the base LoRA's role is to learn general knowledge and assist in subsequent gradient estimation, while the cluster LoRA fine-tunes the model further. Could a higher-rank base LoRA be used, while training lower-rank LoRA for different clusters, to achieve faster adaptation?
2. Regarding clustering by gradients, what patterns exist within the data in each cluster? Are the tasks within each cluster more similar to each other?

---

> ### Author Response · Authors · 2024-11-21
>
> Dear Reviewer,
>
> Thank you very much for your thoughtful and constructive feedback on our work. We sincerely appreciate your support and have addressed your questions and concerns below.
>
> ### Weaknesses
>
> 1.1 **Discussion on the advantages of MoE**: We greatly appreciate your suggestion regarding the advantages of MoE. In response, we have added a discussion on the potential benefits of MoE in the revised Section 2.3, “MoE and Ensembles.” We have also included related experimental results in the revised Section 5, “Results and Discussion” (see two paragraphs starting from Line 371). We hope that this discussion effectively emphasizes the advantages of MoE and strengthens our claims.
>
> 1.2 **Trainable router**: Thank you for this insightful suggestion. Your proposed method resembles the “Mixture of LoRA Experts (MoLE)” method [1] suggested by other reviewers. In the revised Section 5, "Results and Discussion," we have incorporated results from MoLE and provided an in-depth discussion of its performance relative to our model.
>
> 2. **The algorithm’s time complexity**: We appreciate your concern about the algorithm's time complexity. In response, we have provided theoretical analyses and empirical experiments on the resource consumption of ELREA compared to fine-tuning only the base adapter in Appendix G, "Efficiency Analysis," of the revised manuscript. Specifically, for a 4-expert system (in addition to the base adapter), we found that the fine-tuning and inference resource consumption is approximately 2.3 times that of using the base adapter alone. While this increase is not negligible, we believe it is acceptable in practice given the performance improvements. Nevertheless, we acknowledge the importance of efficiency and are actively working on methods to reduce computational costs in our ongoing research.
>
> ### Questions
>
> 1. **Training lower-rank LoRAs on clusters**: Thank you for this practical suggestion. It is indeed possible to use lower-rank LoRAs on the clustered data for faster adaptation. Our main consideration is that, in our experiments, the LoRA experts are initialized from the base adapter, and using lower-rank experts may result in a dimension mismatch. While we can resolve this issue by applying SVD on LoRA weights, implementing this solution requires additional effort. We are committed to exploring this idea in our future work to reduce computational overhead.
>
> 2. **Cluster pattern**: Yes, we have observed a clear correlation of tasks within each cluster. We have added an illustration and analysis in Appendix H “Further Analysis on Data Clustering” of the revised article to provide more insight into this pattern.
>
> We hope that our responses have satisfactorily addressed your questions and concerns. Your valuable feedback has significantly improved our work. If you find our revisions meet your expectations, we kindly hope you might consider reflecting this in your evaluation. Please do not hesitate to reach out if you have any further questions or require additional clarification; we would be delighted to continue the discussion.
>
> Best regards,
> The Authors
>
> [1] Wu, Xun, Shaohan Huang, and Furu Wei. "Mixture of LoRA Experts." The Twelfth International Conference on Learning Representations, 2024, https://openreview.net/forum?id=uWvKBCYh4S.

---

> > ### Comment · Reviewer_Ue2L · 2024-11-23
> >
> > Thank you for your response and the revised manuscript. I have reviewed it carefully, and it has addressed some of my concerns. Based on my overall assessment, I am inclined to keep my initial rating.

---

> ### Author Response · Authors · 2024-11-25
> **Thank you for your response**
>
> Dear Reviewer,
>
> Thank you for reviewing our revised manuscript and for your feedback. We're glad some of your concerns have been addressed. Please let us know if there are specific areas we can improve further.
>
> Best regards,
> The Authors

---

### Official Review · Reviewer_PB5A · 2024-10-31

**Soundness:** 3
**Presentation:** 3
**Contribution:** 2
**Rating:** 6
**Confidence:** 4

**Summary:**

The authors presented a framework aimed at enhancing the fine-tuning of large language models (LLMs) by addressing challenges related to conflicting gradient directions. By clustering training instructions according to their gradient features and independently training expert adapters on these clusters using the Low-Rank Adaptation (LoRA) technique, the authors proposed a method to improve training efficiency and model scalability. The framework dynamically selected the most relevant adapters during inference to optimize performance, which was demonstrated through experiments to outperform baseline methods and other ensemble approaches across various domain-specific tasks.

**Strengths:**

1.	The authors explained the challenges of fine-tuning large models from the perspective of gradient conflicts, which is a novel approach. They then propose clustering training samples based on their gradients and training them separately using LoRA. By integrating multiple LoRA adapters, they enhance the performance of LoRA fine-tuning.

2.	The authors’ writing structure is well-organized and the text is clear and easy to understand. Additionally, the research methodology is rigorous, with detailed experiments conducted.

**Weaknesses:**

1.	The authors did not discuss the complexity of the method. Compared to the vanilla LoRA, this approach introduces a clustering process and the training of multiple LoRAs, which inevitably increases the complexity of training. Additionally, computing different LoRA weight combinations for each sample could also lead to a decline in inference performance. Although the author's method does indeed improve accuracy, whether it is practical in real-world scenarios requires further discussion.

2.	The authors’ experimental comparisons are not sufficiently comprehensive. Currently, there are many methods dedicated to enhancing LoRA's performance on multitasking, such as MixLoRA[1], MultiLoRA[2], and MOELoRA[3]. However, the MOE methods compared by the author appear to be just modified versions of their own approach, which might be more suitable for inclusion in an ablation study. If the authors could compare their method with these more advanced techniques, it would further demonstrate the effectiveness of the proposed method.

[1] Li D, Ma Y, Wang N, et al. Mixlora: Enhancing large language models fine-tuning with lora based mixture of experts[J]. arXiv preprint arXiv:2404.15159, 2024.

[2] Wang Y, Lin Y, Zeng X, et al. Multilora: Democratizing lora for better multi-task learning[J]. arXiv preprint arXiv:2311.11501, 2023.

[3] Liu Q, Wu X, Zhao X, et al. Moelora: An moe-based parameter efficient fine-tuning method for multi-task medical applications[J]. arXiv preprint arXiv:2310.18339, 2023.

3. The authors use the random clustering approach for training, and the performance is only slightly worse than the gradient-based clustering method. This suggests that the gradient-based approach may not be essential, which contradicts the overall motivation of the method.

**Questions:**

1.	Could you have provided more details on the computational overhead associated with the ELREA framework?

2.	You use 5,000 samples for clustering, which is a relatively small portion of the entire training set. If the sample changes, could this lead to different clustering results and significantly alter the final training outcomes? In other words, does this clustering method possess stability?

3.	For the experiments where the performance gains from gradient-based clustering are far less significant than those from LoRA ensembling, can you provide a reasonable explanation? If clustering is not used, and each LoRA is trained using different random seeds and then ensembled, could this also achieve good performance?

---

> ### Author Response · Authors · 2024-11-21
> **Official Comment by Authors [PART I]**
>
> Dear Reviewer,
>
> We sincerely appreciate the time and effort you have dedicated to reviewing our manuscript. Your insightful comments are invaluable to us, and we believe they will greatly enhance the quality of our work. Below, we address your questions and concerns in detail.
>
> ### Weaknesses
>
> 1. **Method complexity**: Thank you for highlighting the need for a detailed discussion on computational overhead. In the revised version of our manuscript, we have included a comprehensive analysis in Appendix G “Efficiency Analysis”. Briefly, we found that ELREA with four expert adapters requires approximately $2.3\times$ the computational resources needed for fine-tuning the base adapter. While this represents an increase, we consider it acceptable given the performance improvements achieved. We are also actively exploring ways to reduce computational overhead in our ongoing research to make our method more practical in real-world scenarios.
>
> 2. **Comparison with baselines**: We are grateful for your suggestions regarding baseline methods. To strengthen our evaluation, we have included results from the Mixture of LoRA Experts (MoLE) [2] in the revised Section 5, "Results and Discussion." MoLE differs from ELREA by fine-tuning additional layer-wise gating functions to dynamically assign weights to expert adapters. We have provided an in-depth analysis of its performance relative to our model.
> Regarding the methods mentioned in references [3,4,5], we acknowledge their relevance and have discussed them in the related work section. These methods differ from ELREA in that they simultaneously train all LoRA experts and gating functions on the entire dataset, which is outside the scope of our current setup. Implementing these baselines would require significant additional time and resources, which may not be feasible within the current discussion period. However, we are committed to including these comparisons in future versions of our work.
>
> 3. **Performance gain**: This point is addressed below.
>
> [1] Fort, Stanislav, Huiyi Hu, and Balaji Lakshminarayanan. "Deep Ensembles: A Loss Landscape Perspective." arXiv preprint arXiv:1912.02757 (2019).
> [2] Wu, Xun, Shaohan Huang, and Furu Wei. "Mixture of LoRA Experts." The Twelfth International Conference on Learning Representations, 2024. https://openreview.net/forum?id=uWvKBCYh4S.
> [3] Li, D., Ma, Y., Wang, N., et al. "MixLoRA: Enhancing Large Language Models Fine-Tuning with LoRA Based Mixture of Experts." arXiv preprint arXiv:2404.15159, 2024.
> [4] Wang, Y., Lin, Y., Zeng, X., et al. "MultiLoRA: Democratizing LoRA for Better Multi-Task Learning." arXiv preprint arXiv:2311.11501, 2023.
> [5] Liu, Q., Wu, X., Zhao, X., et al. "MoELoRA: An MoE-Based Parameter Efficient Fine-Tuning Method for Multi-Task Medical Applications." arXiv preprint arXiv:2310.18339, 2023.

---

> ### Author Response · Authors · 2024-11-21
> **Official Comment by Authors [PART II]**
>
> ### Questions
>
> 1. **Computation overhead**: We have conducted both theoretical and empirical analyses, which are now included in Appendix G "Efficiency Analysis" of the revised manuscript. Specifically, for a 4-expert system (in addition to the base adapter), we found that the fine-tuning and inference resource consumption is approximately 2.3 times that of using the base adapter alone. While this increase is not negligible, we believe it is acceptable in practice given the performance improvements. Nonetheless, we acknowledge that efficiency is crucial, and we are actively working on methods to reduce computational costs in our ongoing research.
>
> 2. **Stability of clustering**: In our preliminary experiments, we observed that changing the random seed for sampling had minimal impact on the clustering results. Specifically, for most random seeds (about 8 out of 10), the distribution of data points among clusters remained consistent, with only about a 5% variation in a few cases. These findings suggest that our clustering method is stable and reliable. We added related content in Section 3.3 “Clustering and Per-Cluster Fine-Tuning”. In addition, we added a set of investigations in the data pattern in each cluster in Appendix H “Further Analysis on Data Clustering”.
>
> 3. **Explanation of Performance Gain**: We understand your concern about the source of the performance gain. We believe you are referring to the comparison between ELREA and the Random Cluster approach when r=8 in Table 1. The Random Cluster method, as detailed in Appendix E, is similar to the Deep Ensembles approach, which has been shown to effectively explore the loss landscape by converging to different minima [1]. Since we use the standard token-wise cross-entropy loss, the training data primarily defines the loss landscape. By training different LoRA adapters that converge to diverse minima, we enhance the ensemble model's representation capability, leading to better performance.
> Our ELREA framework can be seen as a specialized ensemble model that systematically diversifies each LoRA adapter, rather than relying on the stochastic nature of Deep Ensembles. This structured approach aims to provide more consistent performance gains. To further investigate, we conducted additional experiments on the MATH-Combined dataset with four extra LoRA components, each trained on the entire dataset from scratch. Due to computational constraints, we limited the number of LoRAs. The results, now included in the revised Table 1 under "LoRA Ensembles," show that while LoRA Ensembles do provide consistent performance improvements, ELREA achieves a larger gain, aligning with our expectations.
>
> Once again, we thank you for your thoughtful feedback. We hope our responses have addressed your concerns. Your feedback has been invaluable in improving the quality of our work. We hope that our revised manuscript meets the ICLR quality standards and would be grateful if you would consider updating your evaluation accordingly.
>
> Please do not hesitate to let us know if you have any further questions or suggestions. We are more than happy to engage in further discussions to enhance our work.
>
> Best regards,
> The Authors

---

> > ### Comment · Reviewer_PB5A · 2024-11-26
> >
> > I have read the responses of the authors and other reviewers' opinions. However, I am still not convinced by the explanations about the complexity analysis and the explanations about the impact of cluster results. Therefore, I prefer to keep my score. If there are more results about these two problems. I expect to discuss them with the authors.

---

> > > ### Author Response · Authors · 2024-11-26
> > >
> > > Dear Reviewer,
> > >
> > > Thank you for reading our responses and for sharing your thoughts. We would appreciate further guidance on how we might enhance our results and discussion to make them more robust and convincing. Could you please provide more details on the specific results or discussions you believe need improvement, or any weaknesses we should address in our revisions or subsequent discussions?
> > >
> > > Meanwhile, we have updated our manuscript by adding Figure 5 in the appendix, which demonstrates that the data clusters maintain a similar structure across various random seeds, even when the clusters are not identical. We have also included a discussion of this result at the end of Appendix H (Lines 1536--1542). We hope these updates further strengthen our response to your second question [here](https://openreview.net/forum?id=l0gZS0sAlf&noteId=hoGR1lFtPo).
> > >
> > > Regarding the complexity analysis, we would appreciate any suggestions on how we can further strengthen our arguments beyond the results provided in our revised Appendix G "Efficiency Analysis" and Table 5.
> > >
> > > Thank you for your time and valuable feedback!
> > >
> > > Best regards,
> > > The Authors

---

> ### Author Response · Authors · 2024-11-25
> **Reminder to check our response**
>
> Dear Reviewer,
>
> Thank you again for your valuable feedback on our submission. In our response, we tried our best to address your comments and hope our clarifications resolve any concerns. We kindly invite you to review our replies, and let us know if you have further questions. Thank you for your time and consideration.
>
> Best regards,
> The Authors

---

> ### Author Response · Authors · 2024-12-01
> **Please let us know your opinion**
>
> Dear Reviewer,
>
> As the rebuttal session is coming to an end, we kindly ask you to review our [updates and questions](https://openreview.net/forum?id=l0gZS0sAlf&noteId=qOA00enpzi) regarding your response, and let us know how we can further strengthen our arguments beyond the provided results. Thank you for your time and consideration.
>
> Best regards,
> The Authors

---

### Official Review · Reviewer_Wm5H · 2024-11-01

**Soundness:** 3
**Presentation:** 3
**Contribution:** 3
**Rating:** 6
**Confidence:** 2

**Summary:**

This paper proposes a novel two-stage fine-tuning method for large language models, named ELREA. In the first stage, ELREA trains a general adapter on the entire fine-tuning dataset. In the second stage, it clusters the training data into multiple groups and trains a specialized adapter for each group. During inference, ELREA dynamically selects the most suitable adapter based on the gradient similarity between the input data and the training clusters. Experimental results confirm the effectiveness of ELREA.

**Strengths:**

1. The paper is well-written and easy to understand.
2. The combination of LoRA with gradient-based clustering for expert adapters to address the conflicting gradient issue is interesting.
3. The framework’s design is well-structured, and the authors provide extensive experimental validation to demonstrate the efficacy of ELREA.

Since I am not very familiar with the latest work in the field of LLM fine-tuning, I may not be in the best position to assess the novelty of this paper. This paper is interesting for me. I am inclined to accept and respect the opinions of other reviewers.

**Weaknesses:**

1. As discussed by the authors in the limitations section, the computational and memory overhead of ELREA is substantial.
2. The paper lacks an in-depth discussion on the robustness of the clusters and the impact of the clustering algorithm (BIRCH) on performance. Since clustering is central to ELREA, variations in cluster formation could significantly affect adapter performance.

**Questions:**

Why is the base adapter still used during the inference phase? Since the cluster adapters are derived from the base adapter, they should already contain the necessary information from it. Could the authors’ inclusion of the base adapter at inference suggest that the cluster adapters may have forgotten some useful information from other clusters during training?

---

> ### Author Response · Authors · 2024-11-21
>
> Dear Reviewer,
>
> Thank you for your thoughtful review and supportive comments on our work. We are delighted to hear that you found our paper well-written and interesting. Our goal is to make our research accessible even to audiences who may not be deeply familiar with the latest developments in LLM fine-tuning. We sincerely appreciate your feedback and have addressed your concerns below. Please let us know if there's anything else we can do to further enhance the readability and robustness of our work.
>
> ### Weaknesses
>
> 1. **Method efficiency**:  We understand your concern regarding the computational and memory overhead of ELREA. To address this, we have conducted both theoretical and empirical analyses of the method's efficiency, which are now included in Appendix G, "Efficiency Analysis," of the revised manuscript. Specifically, for a system with four expert adapters (in addition to the base adapter), we found that the fine-tuning and inference resource consumption is approximately 2.3 times that of using the base adapter alone. While this increase is notable, we believe it is justified by the significant performance improvements ELREA offers. Nonetheless, we acknowledge the importance of efficiency and are actively exploring ways to reduce computational costs in our ongoing research.
>
>
> 2. **The effect of clustering methods**:  We fully agree that the robustness of the clusters and the choice of clustering algorithm are critical to ELREA's performance. We dedicated considerable effort to selecting a clustering method that balances scalability with robustness. Through several smaller-scale experiments during the early stages of our algorithm's development, we determined that BIRCH is the best-performing clustering method for our purposes. It offers decent scalability with respect to data size and feature dimensionality and demonstrates robustness to outliers. Additionally, we experimented with different random seeds and observed consistent cluster formations, indicating that the clustering method is stable across various scenarios. We have expanded our discussion on this topic in Appendix H “Further Analysis on Data Clustering” to provide a more in-depth analysis on the data patterns in each cluster.
>
> ### Questions
>
>
> 1. **Inclusion of the base adapter**:  Your insight is indeed correct. When we fine-tune a low-rank adapter on a smaller, skill-specific subset, the model can lose some of its generalization capability—specifically, information related to other clusters. To mitigate this, we retain the base adapter during the inference phase to maintain access to general knowledge that may be necessary. We have included a brief discussion on this issue between Equations (9) and (10) in the revised manuscript to clarify this point.
>
>
> We hope that our responses have satisfactorily addressed your questions and concerns. Your valuable feedback has significantly improved our work. If you find our revisions meet your expectations, we kindly hope you might consider reflecting this in your evaluation. Please do not hesitate to reach out if you have any further questions or require additional clarification; we would be delighted to continue the discussion.
>
> Best regards,
> The Authors

---

> ### Author Response · Authors · 2024-11-25
> **Reminder to check our response**
>
> Dear Reviewer,
>
> Thank you again for your valuable feedback on our submission. In our response, we tried our best to address your comments and hope our clarifications resolve any concerns. We kindly invite you to review our replies, and let us know if you have further questions. Thank you for your time and consideration.
>
> Best regards,
> The Authors

---

> ### Comment · Reviewer_Wm5H · 2024-11-26
>
> Thanks for your rebuttal. It has addressed my concerns well. After carefully considering the opinions from the other reviewers, I have decided to maintain my positive score.

---

> ### Author Response · Authors · 2024-11-26
> **Thank you for your feedback**
>
> Dear Reviewer,
>
> Thank you for your thoughtful consideration and positive feedback. We are glad to hear that our rebuttal addressed your concerns effectively. We appreciate your continued support of our work.
>
> Best regards,
> The Authors

---

### Official Review · Reviewer_vSEH · 2024-11-03

**Soundness:** 2
**Presentation:** 3
**Contribution:** 2
**Rating:** 5
**Confidence:** 3

**Summary:**

This paper presents Ensembles of Low-Rank Expert Adapters (ELREA), a framework that combines efficient adaptation techniques to resolve conflicting gradients in LLM fine-tuning. By leveraging gradient clustering, the study creates expert adapters for diverse tasks, independent of specific data features. Results indicate that ELREA outperforms baseline LoRA adapters and other expert and self-consistency methods across various applications.

**Strengths:**

1.  The presentation is well.
2.  The method section of the paper is clearly articulated, especially Figure 1, which is a clear and self-explanatory flowchart.
3.  Major differences of MoE and Deep Ensembles are highlighted (P.4). Assumptions (P.5) and limitations (P.10) are clearly stated.
4.  Included detailed appendices to further explain the datasets, experiment setups so the main body is not bulky.

**Weaknesses:**

Missing Paper Structure. The section “Related Work” should be better put in the Appendix.

Limit of Novelty. This paper proposes combine two points together creatively. However, the method for data selection and partitioning (Section 3.2) directly depends on [1]. Moreover, the per-cluster fine-tuning actually has little to do with LoRA; for instance, we could also use full-parameter or soft-prompt tuning for each cluster. Therefore, the motivation for exclusive “LoRA” is not adequate, which needs further illustration.

Lack of comparison with other “LoRA” baselines. It would be more complete if some mixture-of-lora-expert methods[2,3] are included to compare appropriately.

Some writing typos that will not affect the rating scores:

* A few minor grammatical mistakes are found (e.g. P.4: which is the most adapted for LM fine-tuning; P.6: we the weights do not need to sum to 1).

* Undefined symbols: x_{t} in (1) on P.2; d_{model} on line 2 of P.3; cos’ in (9) on P.6 [they are interpretable but the authors should state clearly].

* Conflicting symbol: sigma^bar_c on the line above (8) and in (8) have different formulae on P.5.

* Figure 1: The paragraph following the section heading “3 Method” (P.4) said “The pipeline, shown in Figure 1, consists of three main steps” but the “steps” are not clearly shown in Figure 1. Simply speaking, the 4 phrases in Figure 1 do not match the 3 steps.

* The reference item “Improving language understanding by generative pre- training” (Radford et al.) on P.15 seems to be incomplete. A possible URL: https://cdn.openai.com/research-covers/language- unsupervised/language_understanding_paper.pdf

[1] LESS: Selecting Influential Data for Targeted Instruction Tuning. 2024 ICML.
[2] HydraLoRA: An Asymmetric LoRA Architecture for Efficient Fine-Tuning. 2024
NeurIPS.
[3] Mixture of LoRA Experts. 2024 ICLR.

**Questions:**

Please see the weakness section.

**Details Of Ethics Concerns:**

N.A.

---

> ### Author Response · Authors · 2024-11-21
>
> Dear Reviewer,
>
> Thank you very much for your careful review of our paper and your insightful suggestions. We are grateful for your feedback, which we believe will significantly improve the quality and robustness of our work.
>
> - **Placement of the “Related Works” section**: We appreciate your suggestion regarding the structure of the paper. Currently, we cited the critical works that our article is based on in the Introduction and Preliminaries and others in the “Related Works” section in Appendix A. Due to length limitations, we may not be able to put the full-fledged related works in the article body.
>
> - **Limit of novelty**: Thank you for sharing your thoughts on our work. We would like to clarify that while we utilize the gradient calculation technique from [1], the rest of our approach, including the overall pipeline, clustering methodology, and inference routing weight calculation, is independently developed. Our work aims to integrate these components in a novel way to address the challenges in LLM fine-tuning. We will make sure to emphasize this distinction more clearly in the revised manuscript to avoid any potential misunderstandings.
>
> - **Per-cluster fine-tuning and LoRA**: We appreciate your thoughts on the relevance of LoRA in per-cluster fine-tuning. Allow us to elaborate on our motivation for exclusively using LoRA adapters. Our primary goal was to maintain computational efficiency comparable to training a single adapter on the entire dataset. Full-parameter fine-tuning for each expert would significantly increase the computational and memory requirements, as it would necessitate storing multiple copies of the full model weights. In contrast, using LoRA adapters allows us to keep the additional memory footprint minimal, as each adapter constitutes only a small fraction (usually 0.1% – 3%) of the backbone model's size. We also added an analysis of ELREA's efficiency in Appendix G "Efficiency Analysis" in the revised manuscript.
> Regarding alternative methods like soft prompt tuning, they often require specific task indicators or prior knowledge of downstream tasks, which may not be applicable in our setup. Therefore, we believe that cluster-wise LoRA fine-tuning is the most practical and efficient approach for our scenario. We are also open to further discussion if you have additional suggestions or insights.
>
> - **Comparison with LoRA baselines**: Thank you for suggesting the inclusion of additional baselines such as mixture-of-LoRA-expert methods [2,3]. We agree that incorporating these comparisons would strengthen our evaluation. In response, we have included results from MoLE [3] in the revised Section 5, "Results and Discussion," and provided an in-depth analysis of its performance relative to our model.
> Regarding HydraLoRA [2], we acknowledge its relevance and have discussed it in the related works section. However, due to differences in its setup compared to ELREA and the time constraints of the rebuttal period, implementing it for a fair comparison may not be feasible at this moment. Nevertheless, we are committed to including such comparisons in future iterations of our work to provide a more comprehensive evaluation.
>
> - **Typo and clarity issues**: Thank you so much for pointing out the minor grammatical mistakes and inconsistencies. We appreciate your attention to these details and have ensured that the revised manuscript reflects these corrections.
>
> Once again, we are grateful for your constructive feedback and believe that your suggestions will help enhance the clarity and impact of our work. We hope that our revised manuscript meets the ICLR quality standards and would be grateful if you would consider updating your evaluation accordingly. Please do not hesitate to let us know if you have any further questions or suggestions. We are more than happy to engage in further discussions to enhance our work.
>
> Best regards,
> The Authors
>
>
> [1] LESS: Selecting Influential Data for Targeted Instruction Tuning. 2024 ICML.
> [2] HydraLoRA: An Asymmetric LoRA Architecture for Efficient Fine-Tuning. 2024 NeurIPS.
> [3] Mixture of LoRA Experts. 2024 ICLR.

---

> ### Author Response · Authors · 2024-11-25
> **Reminder to check our response**
>
> Dear Reviewer,
>
> Thank you again for your valuable feedback on our submission. In our response, we tried our best to address your comments and hope our clarifications resolve any concerns. We kindly invite you to review our replies, and let us know if you have further questions. Thank you for your time and consideration.
>
> Best regards,
> The Authors

---

> > ### Author Response · Authors · 2024-11-27
> > **Reminder to check our response**
> >
> > Dear Reviewer,
> >
> > As we approach the final stage for revisions, we are eager to ensure that our responses have fully addressed your questions and concerns. We would be happy to hear from you and address any additional suggestions you might have to enhance our manuscript further. Thank you again for your time and effort in reviewing our paper and your valuable feedback!
> >
> > Best regards,
> > The Authors

---

> > > ### Author Response · Authors · 2024-12-01
> > > **Reminder to check our response**
> > >
> > > Dear Reviewer,
> > >
> > > As the rebuttal session is coming to an end, we kindly ask you to review our [overall response](https://openreview.net/forum?id=l0gZS0sAlf&noteId=vnAlEBEWFl) and our [previous reply](https://openreview.net/forum?id=l0gZS0sAlf&noteId=QIT1HjrXOB) addressing your questions and concerns. Please let us know if you have any additional questions or feedback.
> > >
> > > Thank you for your time and consideration.
> > >
> > > Best regards,
> > > The Authors

---

### Author Response · Authors · 2024-11-22
**Overall response to all reviewers**

Dear Reviewers,

We sincerely thank you for the time and effort you have dedicated to reviewing our manuscript and for providing thoughtful and constructive feedback. Your valuable suggestions have been instrumental in improving the quality of our work.

In this overall response, we summarize the changes we have made to address your general concerns, presented in order of their importance. For your convenience, we have also included a PDF highlighting the updates made in the revised manuscript as **supplementary material**.

1. **Baselines**:  We have incorporated MoLE [1] (trainable gating functions) and LoRA Ensembles [2] as additional baseline methods in our study. The corresponding results and discussions have been updated in the manuscript, primarily in Section 4, Section 5 (Tables 1 & 2), and Appendix E.


2. **Method efficiency**: To address concerns about efficiency, we have added Appendix G, titled "Efficiency Analysis," which includes Table 5. This section provides a comprehensive discussion of efficiency from both theoretical and empirical perspectives. Overall, we find the computational overhead to be reasonable given the performance improvements achieved.


3. **Cluster patterns**: To further clarify data clustering behavior, we have added Appendix H, titled "Further Analysis on Data Clustering," which includes Figure 4 to illustrate data patterns within clusters. Additionally, we have included a sentence in Section 3.3 to confirm that these patterns are robust to variations in random seeds.


4. **Top-k experts**: We have added Figure 3(b) to illustrate the performance when only the Top-1 or Top-3 experts are activated during inference, along with a new paragraph (Lines 502–506) providing a detailed analysis of the results.


In addition to this overall response, we have provided explanations and addressed specific points in our individual responses to each reviewer. We hope that our revisions and responses adequately address your concerns, and we welcome any additional feedback or questions you may have.

Best regards,
The Authors

[1] Wu, Xun, Shaohan Huang, and Furu Wei. "Mixture of lora experts." arXiv preprint arXiv:2404.13628 (2024).
[2] Wang, Xi, Laurence Aitchison, and Maja Rudolph. "LoRA ensembles for large language model fine-tuning." arXiv preprint arXiv:2310.00035 (2023).

---

### Meta-Review · Area_Chair_Z9rg · 2024-12-17

**Metareview:**

In this paper, the authors proposed a new framework for fine-tuning LLMs, where training data is first clustered into different groups based on a global adaptor with the LoRA technique, different expert adaptors are then trained for each group, and finally, an ensemble model is applied to generate the final fine-tuning outcomes.

The idea of the proposed framework is interesting, and the design is reasonable. The performance of the proposed framework is promising. This is a borderline paper, As most of the concerns raised by the reviewers about the complexity, novelty, and comparable experiments of the proposed framework have been addressed during the rebuttal, I recommend acceptance of this paper.

**Additional Comments On Reviewer Discussion:**

Most of the concerns have been addressed during the rebuttal.

---

### Decision · Program_Chairs · 2025-01-22

Accept (Poster)